# Adrenergic modulation of melanocortin pathway by hunger signals

Nilufer Sayar-Atasoy [1,6], Connor Laule[1,6], Iltan Aklan [1], Hyojin Kim[1], Yavuz Yavuz[2], Tayfun Ates[1], Ilknur Coban [3], Fulya Koksalar-Alkan [4], Jacob Rysted[1], Debbie Davis[1], Uday Singh[1], Muhammed Ikbal Alp[5], Bayram Yilmaz[2], Huxing Cui [1] & Deniz Atasoy [1] ✉

Norepinephrine (NE) is a well-known appetite regulator, and the nor/adrenergic system is targeted by several anti-obesity drugs. To better understand the circuitry underlying adrenergic appetite control, here we investigated the paraventricular hypothalamic nucleus (PVN), a key brain region that integrates energy signals and receives dense nor/adrenergic input, using a mouse model. We found that PVN NE level increases with signals of energy deficit and decreases with food access. This pattern is recapitulated by the innervating catecholaminergic axon terminals originating from NTS[TH]-neurons. Optogenetic activation of rostral-NTS[TH] → PVN projection elicited strong motivation to eat comparable to overnight fasting whereas its inhibition attenuated both fasting-induced & hypoglycemic feeding. We found that NTS[TH]-axons functionally targeted PVN[MC4R]-neurons by predominantly inhibiting them, in part, through α1-AR mediated potentiation of GABA release from ARC[AgRP] presynaptic terminals. Furthermore, glucoprivation suppressed PVN[MC4R] activity, which was required for hypoglycemic feeding response. These results define an ascending nor/adrenergic circuit, NTS[TH] → PVN[MC4R], that conveys peripheral hunger signals to melanocortin pathway.

Obesity is a major public health problem with increasing prevalence over the last few decades[1]. In addition to reducing overall quality of life, obesity-associated comorbidities constitute a significant burden on overall healthcare system. Ever increasing need for pharmacologic approaches to treat obesity makes it crucial to understand circuits underlying energy homeostasis. Norepinephrine (NE) has long been known to robustly regulate appetite[2] and some of the current anti-obesity drugs are thought to recruit nor/adrenergic system as a secondary effect, that may contribute their primary anorexigenic mechanisms[3]. Understanding nor/adrenergic regulation of appetite is also important to gain insight into the mechanism of how other

therapeutics targeting nor/adrenergic system, such as those used for mood disorders, affect appetite and weight gain.

The brainstem is generally known to relay signals of satiety whereas hypothalamic arcuate nucleus (ARC) harbors both hunger promoting and suppressing neurons[4–6]. The interoceptive information from these regions is integrated in paraventricular hypothalamic nucleus (PVN) and other centers to maintain energy homeostasis. Ablating nor/adrenergic inputs to the PVN, which mainly originates from hindbrain catecholamine neurons, impairs feeding in response to hypoglycemia, suggesting that the brainstem may also convey orexigenic signals[7,8]. Likewise, direct norepinephrine (NE) injections into the

[1]Department of Pharmacology, Iowa Neuroscience Institute, Roy J. and Lucille A. Carver College of Medicine, University of Iowa, Iowa City, IA, USA. [2]Department of Physiology, School of Medicine, Yeditepe University, Istanbul, Turkey. [3]Institute for Anatomy and Cell Biology, Heidelberg University, Heidelberg, Germany. [4]Department of Oncology, Wayne State University, Detroit, USA. [5]Department of Physiology, School of Medicine, Research Institute for Health Sciences and Technologies (SABITA), Istanbul Medipol University, Istanbul, Turkey. [6]These authors contributed equally: Nilufer Sayar-Atasoy, Connor Laule. ✉e-mail: deniz-atasoy@uiowa.edu

PVN can potently stimulate feeding behavior[9]. Consistent with an appetite regulating role of nor/adrenergic input to PVN, earlier microdialysis experiments suggested dynamic regulation of PVN NE and epinephrine (E) levels in coordination with energy demands[10,11]. Further studies using pharmacologic blockers provided initial insights into the intricacies of this nor/adrenergic input; depending on the targeted receptor subtype, where α1 and α2 receptor agonists decreased and increased feeding, respectively[12,13]. These studies highlight the complexity of NE/E signaling in appetite regulation[2]. However, due to their limited specificity, mechanisms of how hindbrain catecholamine neurons communicate peripheral signals of energy deficit to modulate appetite remain unresolved. Furthermore, the precise identity of neurons providing feeding relevant NE/E and their downstream effectors in the PVN are poorly understood.

Recently, a subset of NTS (nucleus of the solitary tract) catecholamine neurons were shown to receive vagal input and have the capacity to promote and suppress feeding[14–19]. Given their extensive projections to the PVN, we hypothesized that catecholaminergic NTS → PVN projections convey energy deprivation signals and modulate appetite. Here, we investigated the role of this circuit in feeding using in vivo fiber photometry recording, optogenetic and chemogenetic behavioral analysis, and ex vivo electrophysiology. We found that PVN NE/E levels and NTS[TH] (tyrosine hydroxylase) neuronal input to PVN are rapidly modulated by feeding and hypoglycemia. NTS[TH] → PVN projections are necessary and sufficient for food deprivation and glucoprivation induced feeding. Lastly, we identified MC4R expressing neurons in the PVN as one of the key targets whose suppression is necessary for glucoprivic appetite. These results reveal a circuit by which visceral information related to energy deficit is conveyed by ascending brainstem fibers and integrated within the hypothalamic melanocortin pathway during fasting-induced feeding and glucose counterregulation.

## Results

### Dynamic modulation of PVN nor/epinephrine levels by feeding and glucoprivation

To gain insight into the role of PVN catecholamine signaling in appetite regulation, we measured NE/E release during feeding with sub-second temporal resolution using the fluorescent GRAB_NE sensor[20] (Fig. 1a). Cre-dependent *AAV-DIO-GRAB_NE2h* and an optical fiber were targeted to the PVN of *Sim1-cre* mice for selective measurement of NE/E release onto PVN neurons (Fig. 1b, c). In fasted mice, presentation of chow food, but not an inedible, known control object (an empty 1.5 ml Eppendorf tube), evoked biphasic NE/E dynamics in the PVN. Food access initially elicited rapid NE/E release followed by a sustained decrease (Fig. 1d–f). The initial rise in NE/E signal was likely mediated by food-related sensory cues since it was also observed in fasted mice given inaccessible food (Supplementary Fig. 1a–d). This response was specific to fasted state since there was no significant difference observed between chow and object presentation in *ad libitum* fed mice (Supplementary Fig. 1e–h). Chronic fiber photometry recordings revealed that NE/E signal edged toward higher level after 18 h in the absence of food compared to the free feeding state in the same animals (Supplementary Fig. 2a–c). Consistent with a role in appetite regulation, intraperitoneal administration of metabolic signals related to hunger such as ghrelin increased PVN NE/E levels (Fig. 1g, h). A key hallmark of hunger state is increased AgRP neuron activity. Given that much of the hunger state physiology and behavior can be mimicked by artificial activation of AgRP neurons, we asked whether this activity could also lead to increased NE/E release in the PVN. To test this, we expressed hM3Dq activating DREADD in AgRP neurons of *Agrp-ires-cre* mice in which PVN NE/E levels were simultaneously measured by GRAB_NE2h sensor. We found that chemogenetic activation of AgRP neurons in sated mice steadily increased PVN NE/E signals (Supplementary Fig. 2g–k), suggesting that hunger-mediated augmentation of

nor/adrenergic input to PVN might, in part, be mediated by AgRP neurons.

Based on lesion experiments implicating PVN NE/E signaling in glucoprivic feeding[21,22], we next assessed the effect of a 2DG-induced hypoglycemic like state on PVN NE/E levels. Consistently, 2DG-induced glucoprivation evoked robust GRAB_NE2h signal in the PVN relative to control injection (saline), suggesting NE/E release (Fig. 1g, h). Collectively, these findings are consistent with modulation of intra-PVN NE/E dynamics by energy deficit and feeding.

### Activity of NTS[TH]→PVN projection is modulated by feeding and glucoprivation

We next sought to identify the source of nor/adrenergic input that conveys feeding and hypoglycemia-related information to the PVN. The NTS integrates peripheral signals and broadcasts this information throughout the brain to regulate metabolic homeostasis. Catecholaminergic NTS neurons and their projections to the PVN are implicated in appetite and physiological stress[23–26]. Alternatively, locus coeruleus (LC) neurons also send abundant NE/E fibers to the hypothalamus and are implicated in satiety[27]. To determine whether NTS or LC is the source of nutritional state-dependent PVN NE/E signaling, we first targeted *AAV-FLEX-tdTomato* to these regions of *Th-ires-cre* mice to visualize projections in the PVN (Fig. 2a). TH immunostaining confirmed tdTomato expression specifically in catecholaminergic neuron soma (Fig. 2b and Supplementary Fig. 3a). Robust NTS[TH] innervation throughout the PVN confirmed earlier studies and provides a neuroanatomical basis for this connection whereas LC[TH] axons had sparse labeling in the PVN; therefore, we focused on NTS projections (Fig. 2c and Supplementary Fig. 3a, b). To assess whether NTS[TH] → PVN projection conveys feeding and glucoprivation related signals, NTS[TH] neurons were selectively transduced with Axon-GCaMP6s and an optical fiber was implanted over the PVN to image activity dynamics of this connection in awake behaving mice (Fig. 2d). Consistent with the PVN NE/E release patterns observed with GRAB_NE2h sensor, presenting food evoked a biphasic NTS[TH] → PVN response with an initial increase (-10 min) followed by silencing in fasted mice (Fig. 2e–g) but not in free feeding mice (Supplementary Fig. 1i–l), suggesting that NTS[TH] might be a major source of NE release in the PVN during feeding. Similar to NE/E release, NTS[TH] → PVN activity also increased with 18 h of fasting (Supplementary Fig. 2d–f).

We previously showed that NTS[TH] neurons orchestrate glucoprivic feeding, in part, through projections in the ARC[14]. However, local silencing of these projections did not completely abolish 2DG induced feeding response, suggesting parallel pathways are also recruited. We next tested whether NTS[TH] → PVN activity is also recruited by hypoglycemia. Induction of a hypoglycemic-like state with 2DG profoundly increased NTS[TH] → PVN axon activity in awake behaving mice (Fig. 2h–j), consistent with 2DG-induced NE/E release in the PVN. However, unlike NE/E signal, NTS[TH] → PVN activity briefly dropped below baseline in the first minutes after 2DG injection. Additionally, we noticed that NE/E based GRAB_NE2h signal slightly precedes NTS[TH] axonal Ca[2+] signal. It is possible that lack of a corresponding drop in NE/E signal and temporal lag between the two signals might be due to adrenergic input from areas other than NTS which may be rapidly activated by 2DG. These findings reveal that NTS[TH] → PVN axon activity mirrors PVN GRAB_NE2h signal in response to food and glucoprivation; thus, these fibers might contribute to metabolic state-dependent dynamics in PVN NE/E levels.

### NTS[TH]→PVN activity is sufficient to promote feeding and corticosterone release

Given the metabolic state sensitive dynamics of PVN NE/E release and NTS[TH] → PVN projection activity, we next asked whether the NTS[TH] → PVN connection has a causal role on appetite regulation using projection-specific optogenetics. NTS[TH] neurons were transduced with the excitatory opsin, ChR2-eYFP, and a fiber was placed over the PVN.

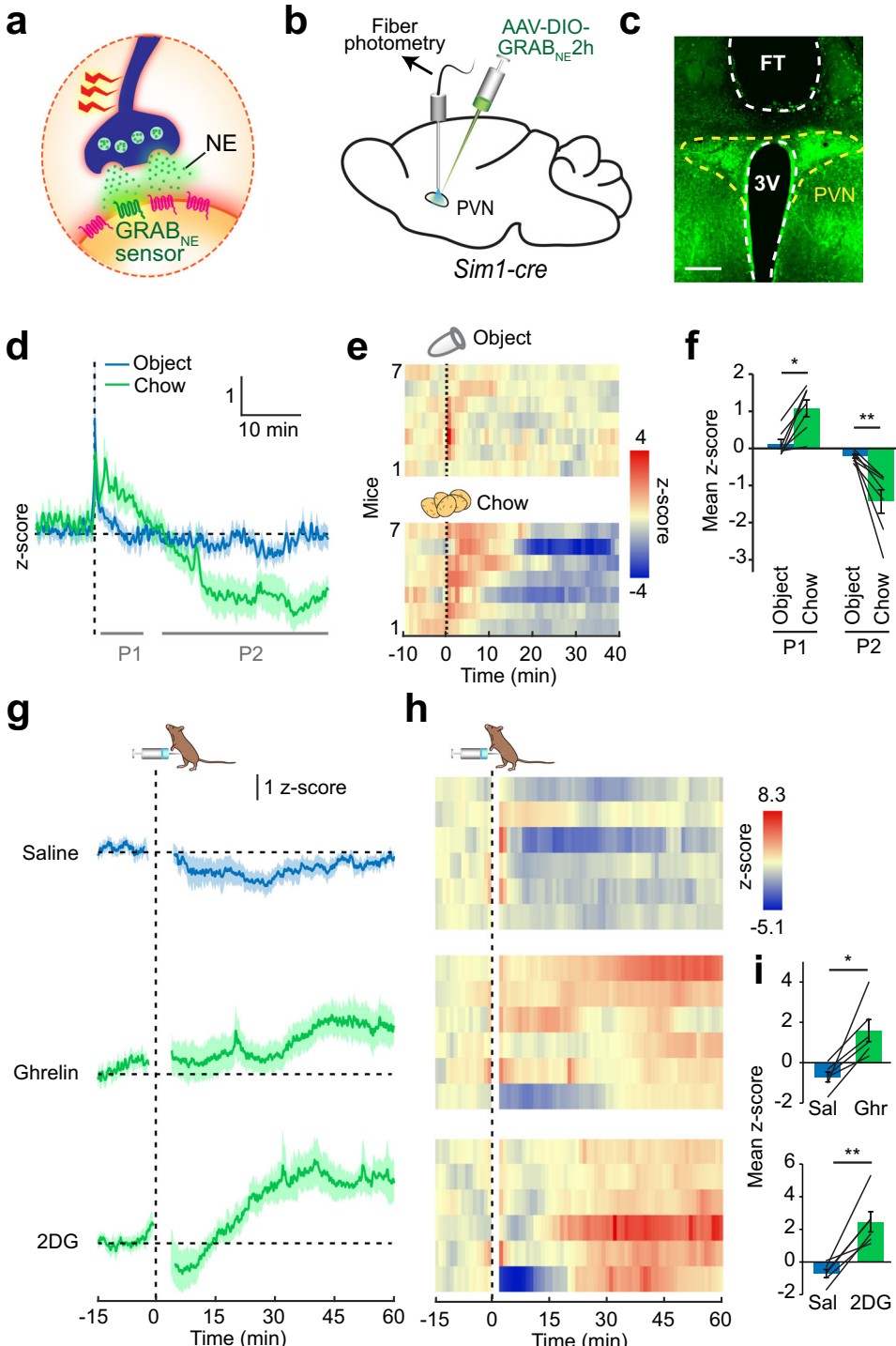

**Fig. 1 | Nor/epinephrine release in the PVN is sensitive to feeding and hunger signals. a** Schematic of fluorescent NE/E reporter, GRAB$_{NE}$2h. **b** GRAB$_{NE}$2h was selectively expressed in Sim1 neurons, and an optical fiber was placed over the PVN to monitor in vivo NE/E dynamics. **c** Representative photomicrograph showing fiber tip (FT) placement over PVN neurons expressing GRAB$_{NE}$2h sensor. Scale: 200 μm. **d** Average GRAB$_{NE}$2h fluorescent response following access to inedible control object or food in fasted mice. **e** Temporal heatmap of response from individual mice and **f** quantification during initial (segment P1) and prolonged

(segment P2) access to object and food ($n = 7$ mice, 2-way RM ANOVA with Tukey's multiple comparison, *$p = 0.023$, **$p = 0.0076$). **g–i** PVN GRAB$_{NE}$2h response to intraperitoneal administration of saline (control), ghrelin (0.5 mg/kg) and 2DG (450 mg/kg), shown as average response (**g**), temporal heatmap from individual mice (**h**), and quantification of mean z-scores (**i**, ghrelin and 2DG: 20–60 min; $n = 6$ mice, two-tailed, paired t-test). *$p = 0.010$, **$p = 0.0035$. Data are presented as mean values +/− SEM.

Optogenetic activation of NTS$^{TH}$ → PVN neurons was validated by high levels of eYFP and cFos colocalization in NTS$^{TH}$ somas and slice recordings (Fig. 3a, e upper panel). Stimulating NTS$^{TH}$ → PVN fibers evoked robust food intake relative to tdTomato injected controls

during both light and dark cycles (Fig. 3b, c and Supplementary Fig. 4c). Food intake positively correlated with the number of ChR2 expressing NTS$^{TH}$ neurons and stimulation frequency, which saturated at 10 Hz (Fig. 3d, e). Consistent with increased consumption during

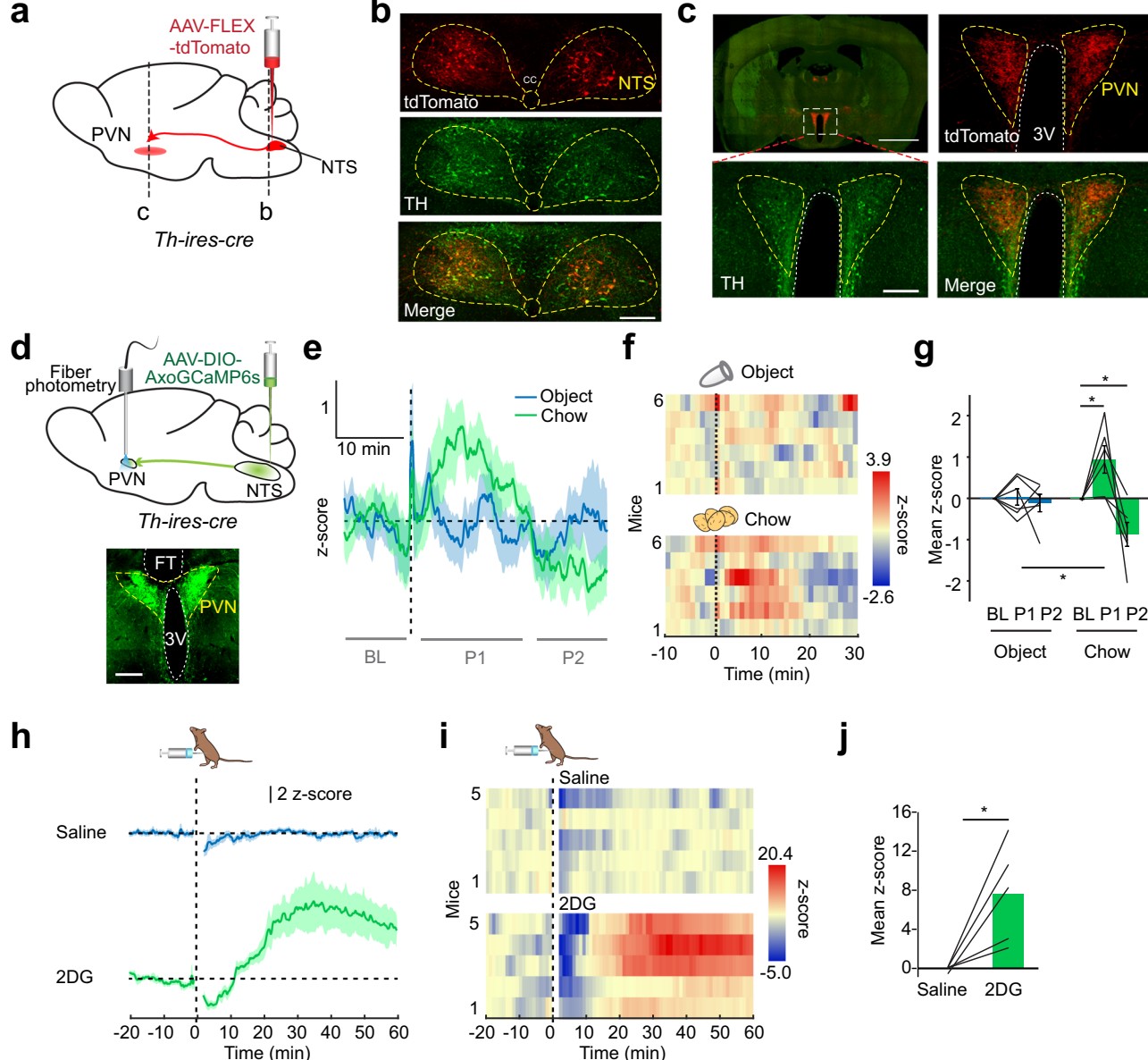

**Fig. 2 | NTS^TH → PVN projection responds to feeding and glucoprivation.**
**a** Schematic for mapping NTS^TH projections in the PVN. **b** Photomicrographs showing fluorescent reporter, tdTomato (red) and anti-TH stained (green) NTS somas. Scale bar: 100 μm. **c** Visualization of PVN innervation by TH positive (green) NTS^TH fibers (red). Note that in addition to incoming TH⁺ axons, local TH-expressing soma in PVN is also labeled with TH staining. Scale bars: 2 mm and 200 μm. **d** Schematic for fiber photometry recording of NTS^TH → PVN axonal activity with Axon-GCaMP6s. Scale bar: 200 μm. **e–g** NTS^TH:AxoGCaMP6s → PVN response to control object and chow in food restricted animals. **e** average traces, **f** individual temporal heatmaps, and **g** quantification at indicated timepoints ($n = 6$ mice, 2-way RM ANOVA with Tukey's and Sidak's multiple comparisons, *$p < 0.05$). **h–j** NTS^TH:AxoGCaMP6s → PVN fluorescent response to intraperitoneal saline and 2DG (450 mg/kg) administration, **h** average traces, **i** individual temporal heatmaps, and **j** quantification of mean z-scores (15–60 min, $n = 5$ mice, two-tailed, paired t-test, *$p = 0.028$). Data are presented as mean values +/− SEM. For specific data, p-values and statistics, please see associated Source data file.

NTS^TH → PVN stimulation, mice displayed more entries and spent significantly more time at food area (FA) with a drastic reduction in latency to enter FA (Fig. 3f–i). Notably, activation of this circuit significantly increased the frequency of meal bouts, but not meal size, suggesting a role for this pathway in meal initiation (Supplementary Fig. 4a, b).

Based on recently reported functional heterogeneity along the rostro-caudal NTS^TH neurons in regulating feeding[15], we selectively targeted rostral NTS^TH neurons (rNTS^TH) to evaluate whether their projections to PVN is sufficient to promote food intake. We found that optogenetic activation of rNTS^TH axons over PVN caused a rapid and strong feeding response (Fig. 3j–m). Notably, latency for first pellet was significantly shorter in rNTS^TH compared to stimulating entire

NTS^TH fibers, consistent with anorexigenic impact of caudal NTS^TH (cNTS^TH) neurons (Fig. 3n). Furthermore, 30 min activation of rNTS^TH → PVN connection was still capable of eliciting significant feeding response even 30 min after stimulation was ceased (Fig. 3o, p), suggesting that continued activity of these neurons is not required and orexigenic drive can linger into post-activation period akin to NPY signaling from ARC^AgRP neurons[28]. Finally, we measured whether rNTS^TH activation can also increase motivation to work for food. Using nose poke based progressive ratio task, we found that rNTS^TH-induced feeding response was indeed accompanied by increased motivation to acquire food comparable to fasting (Fig. 3q–s). Conversely, activating caudal-NTS^TH → PVN connection did not elevate feeding in light phase nor caused feeding suppression in dark phase (Supplementary Fig. 5).

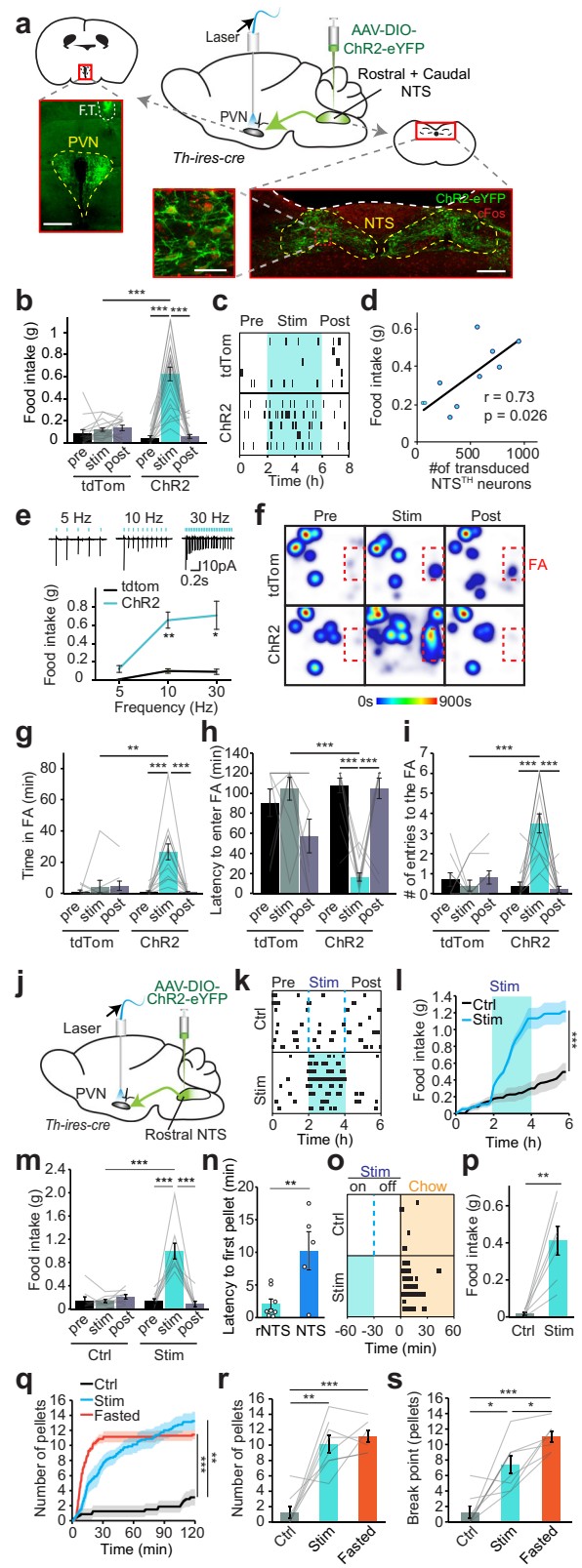

**Fig. 3 | NTS^TH → PVN projection evokes feeding. a** Schematic and representative images of ChR2 expression in NTS^TH-neurons of *Th-ires-cre* mice, showing PVN innervation by NTS^TH:ChR2 axons (top left, F.T.: fiber tract, scale bar 300 μm) and transduced TH-neurons expressing ChR2 (green) and cFos (red) in NTS (bottom, scale bars: 300 μm and 50 μm). **b** Average 2-h food intake during baseline (pre), optogenetic stimulation (stim), and post-stimulation (post) periods in control (tdTomato, $n = 15$) and ChR2 transduced mice ($n = 23$, 2-way RM-ANOVA, ***$p < 0.0001$). **c** Temporal raster plot of food pellets consumed during experiment in tdTomato and ChR2 expressing mice ($n = 5$ each). **d** Two-tailed Pearson correlation between number of ChR2 transduced NTS^TH-neurons and food consumption ($n = 10$). **e** Top: Representative cell-attached recordings from NTS^TH-neurons with 5, 10, and 30 Hz optogenetic stimulations in acutely prepared slices. Bottom: Food intake during NTS^TH → PVN photoactivation at different stimulation frequencies ($n = 5$ tdTomato, $n = 7$ ChR2 mice, 2-way RM-ANOVA, *$p = 0.025$, **$p = 0.0022$). **f** Cumulative heat maps depicting time spent in different locations of the behavioral cage during NTS^TH → PVN photostimulation (FA: food area). **g–i** Quantification of (**g**) time spent in, (**h**) latency to enter, and (**i**) number of entries to the FA in tdTomato ($n = 10$) and ChR2 ($n = 14$) cohorts (2-way RM-ANOVA, ***$p < 0.0001$, **$p = 0.0066$). **j** Schematic for selective photoactivation of rostral NTS^TH (rNTS^TH) neuronal axons over PVN. **k–m** Temporal raster plot (**k**), cumulative (**l**, $n = 9$, two-tailed, paired t-test, ***$p = 0.0008$) and total average feeding (**m**, $n = 9$ mice, 2-way RM-ANOVA, ***$p < 0.0001$) in response to 2-h rNTS^TH → PVN photostimulation during light cycle. **n** Summary graph showing latency to first pellet in response to rNTS^TH → PVN versus NTS^TH → PVN photostimulation ($n = 5$ NTS, 8 rNTS, two-tailed, unpaired t-test, **$p = 0.0067$). **o, p** Raster plot (**o**) and summary graph (**p**) showing delayed feeding 30 min after rNTS^TH → PVN photostimulation is over (stimulation duration: 30 min, control: no stimulation $n = 7$ mice, two-tailed paired t-test, **$p = 0.0022$). **q–s** Progressive ratio task by nose-poke induced pellet delivery in response to 2-h rNTS^TH → PVN potostimulation. Cumulative graph showing acquired pellets (**q**) under photostimulated (stim), non-stimulated (ctrl) and fasted conditions. Average number of pellets earned in 1 h (**r**) and 10 min nose-poke break points (**s**, $n = 8$ ctrl and stim, $n = 7$ fasted mice, One-way ANOVA, *$p < 0.032$, **$p < 0.0033$, ***$p < 0.001$). Data are presented as mean values +/− SEM. For specific $p$-values, data and statistics, please see associated Source data file.

significant change in plasma glucose levels (Supplementary Fig. 4e–g). This confirms that NTS^TH neurons are an upstream circuit node in the HPA axis involved in stress response and hypoglycemia counter-regulation. Notably, cort release was not required for the feeding effect since systemic administration of glucocorticoid receptor antagonist mifepristone did not affect NTS^TH → PVN mediated feeding (Supplementary Fig. 4d); however, we cannot rule out possible contribution of sympathetic activation.

## PVN projecting NTS^TH neurons partake in fasting induced and hypoglycemic feeding

Based on our in vivo fiber photometry recording results suggesting that fasting and hypoglycemia promote NE/E release from NTS^TH axons in the PVN, we next asked whether this projection is required for deprivation- and hypoglycemic-induced hunger. To test this, we employed an intersectional chemogenetic approach to selectively silence the subset of NTS^TH neurons that project to the PVN (Fig. 4a). We injected *rgAAV-pEF1a-DIO-FLPo* into the PVN of *Th-ires-cre* mice. This vector is retrogradely transported from axon terminals back to soma in NTS and expresses FLPo-recombinase in a Cre-dependent manner in TH⁺ neurons. A second virus, *pAAV-hSyn-fDIO-hM4Di-mCherry*, expressing flp-dependent hM4Di was delivered to the NTS (Fig. 4a). Chemogenetic inhibition of NTS^TH → PVN neurons significantly attenuated food consumption after 18 h food deprivation, consistent with the recruitment of this pathway during homeostatic hunger (Fig. 4b). Moreover, 2DG-induced food intake was also significantly suppressed, suggesting a role for NTS^TH → PVN projections in hypoglycemic hunger (Fig. 4c). These findings reveal the necessity for PVN projecting NTS^TH neurons in fasting- and glucoprivation-induced feeding.

Based on the recent finding that catecholaminergic neurons in the rostral and caudal NTS may have opposing effects on appetite, we

Collectively, our results establish an orexigenic role for the rNTS^TH → PVN connection.

Based on the critical role for catecholaminergic NTS neurons and the PVN in stress and hypoglycemia response through the HPA axis, we also investigated the effects of NTS^TH → PVN circuit in glucose homeostasis. As expected, NTS^TH → PVN stimulation increased systemic corticosterone (cort) levels; however, we did not observe a

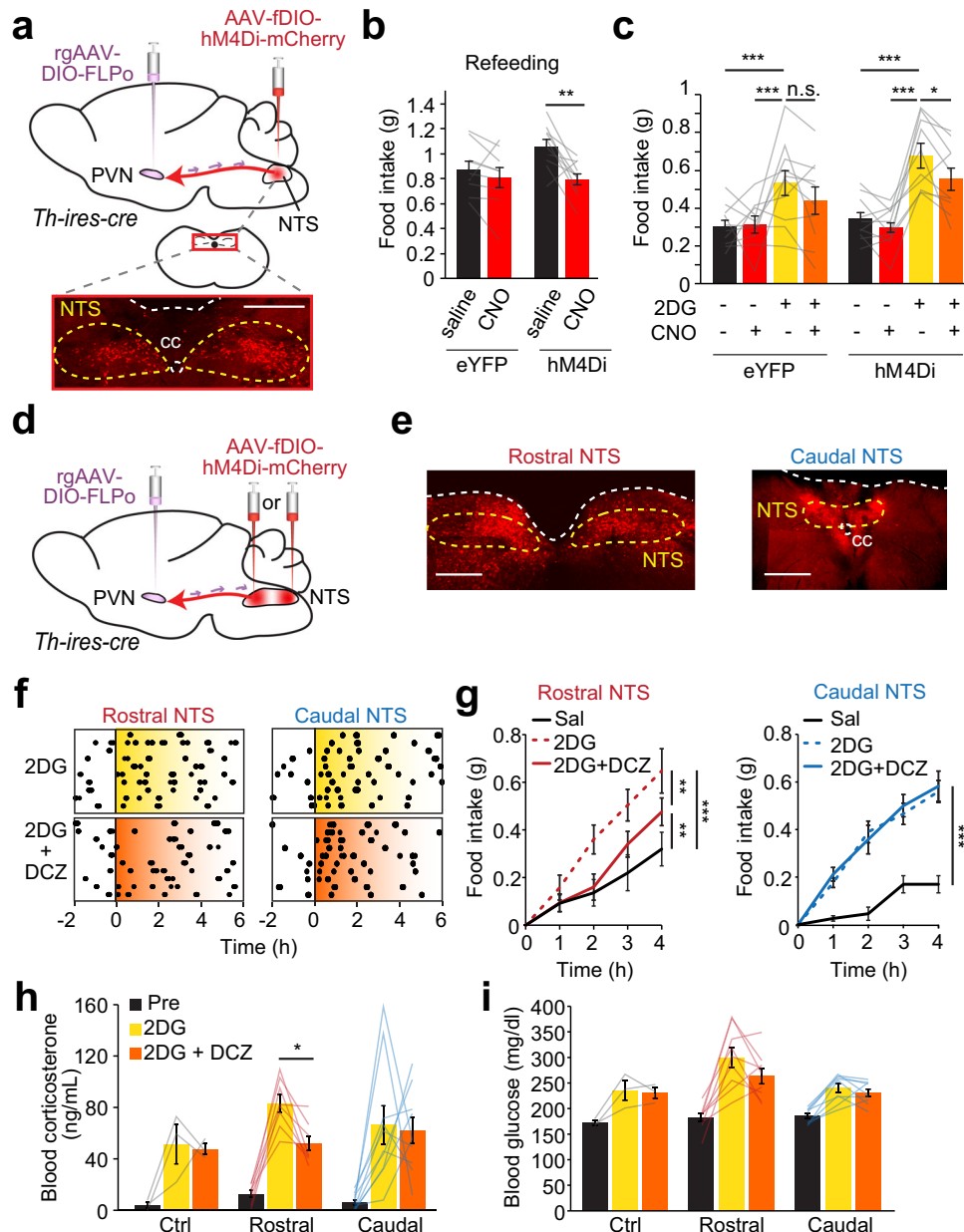

**Fig. 4 | NTS^TH → PVN connection contributes to fasting-induced and glucoprivic feeding. a** Injection diagram for hM4Di-mediated (red) inhibition of PVN-projecting NTS^TH neurons and photomicrograph showing transduced NTS neurons (scale: 500 μm). **b** Effect of control (saline) injection and chemogenetic (CNO) NTS^TH → PVN silencing on 1-h food consumption following overnight food deprivation in control eYFP (n = 9 mice) and hM4Di (n = 10 mice) animals (2-way RM ANOVA with Sidak's multiple comparison, **p = 0.0014). **c** Effect of NTS^TH → PVN inhibition on 2DG-induced feeding in *ad libitum* fed control eYFP (n = 10) and hM4Di (n = 10) mice (4-h feeding, 2-way RM-ANOVA with Benjamini-Hochberg correction, *p = 0.035, ***p < 0.0007, n.s.: p = 0.112, Effect sizes: D = 0.639 (Δ (2DG − Baseline) of Control vs Rostral; D = 0.813 (Δ (2DG:CNO − 2DG) of Control vs Rostral). **d** Schematic for hM4Di transduction in PVN-projecting TH neurons in the rostral or caudal NTS. **e** Representative image of hM4Di (red) expression in PVN-projecting catecholaminergic neurons in the rostral and caudal NTS (scale: 500 μm). **f** Temporal raster plots of food pellets dispensed before and after

intraperitoneal 2DG or 2DG + DCZ administration from mice expressing hM4Di in rostral (n = 10 mice) or caudal (n = 10 mice) NTS^TH → PVN neurons. **g** Summary graph for cumulative 4-h food intake following intraperitoneal saline, 2DG, and 2DG + DCZ delivery from mice expressing hM4Di in PVN-projecting rostral (n = 10 mice) or caudal (n = 10 mice) NTS^TH neurons (2-way RM-ANOVA with Tukey's multiple comparison, ***p < 0.0001, **p < 0.0039). **h, i** Effects of 2DG alone or 2DG + DCZ on circulating corticosterone (**h**) and glucose (**i**) in control (n = 3 mice) and animals with hM4Di expression in rostral (n = 8 mice) or caudal (n = 10 mice) NTS^TH → PVN neurons (One-way RM-ANOVA with Tukey's multiple comparison, *p = 0.0159, Effect sizes: D = 0.987 (Δ (2DG − Baseline) of Control vs Rostral; D = 0.977 (Δ (2DG:DCZ − 2DG) of Control vs Rostral; Effect sizes: D = 0.291 (Δ (2DG-Baseline) of Caudal vs Rostral; D = 0.490 (Δ (2DG:DCZ − 2DG) of Caudal vs Rostral). Data are presented as mean values +/− SEM. For specific p-values, data and statistics, please see associated Source data file.

sought to selectively determine the role of these PVN projecting catecholaminergic NTS subsets in hypoglycemic feeding[15]. We employed the same intersectional approach as above but with specific targeting of fDIO-hM4Di to rostral or caudal regions of the NTS (Fig. 4d, e and Supplementary Fig. 6). We found that selective

inhibition of PVN projecting rostral, but not caudal, NTS^TH neurons significantly reduced 2DG-induced food consumption (Fig. 4f, g).

Given our finding that NTS^TH → PVN activation increases corticosterone levels, we next asked if this projection contributes to hypoglycemia counterregulation and if this is mediated by rostral or caudal

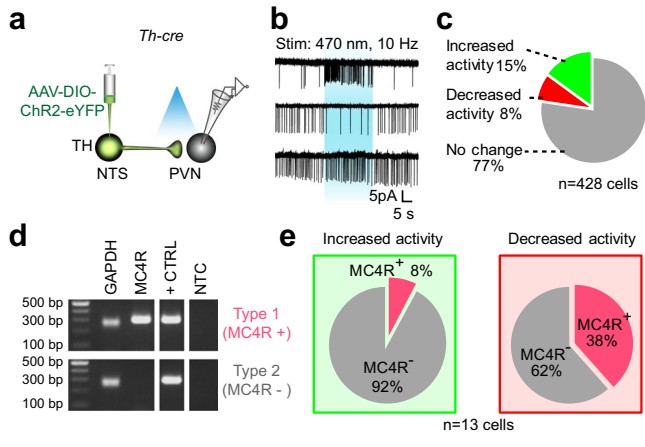

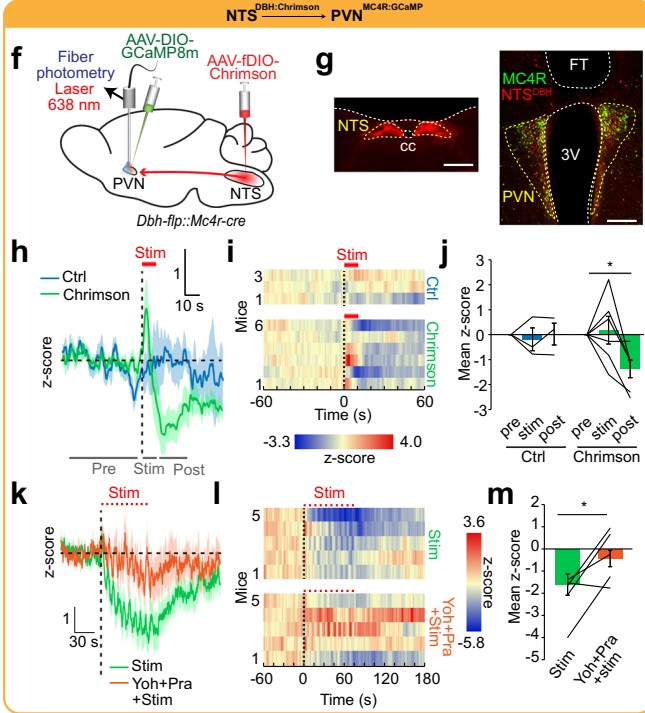

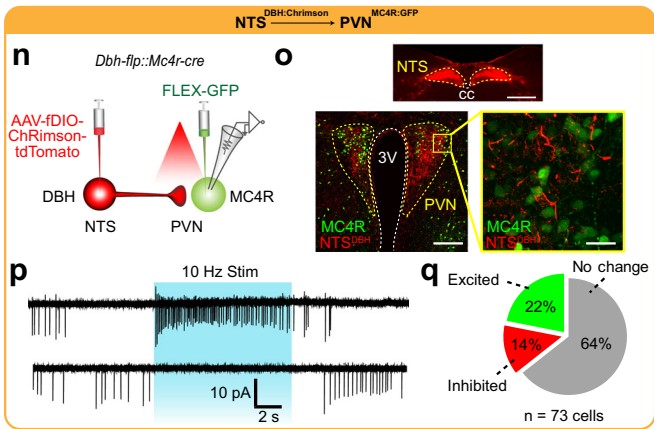

**Fig. 5 | Activating NTS^TH → PVN connection produces heterogenous response profile in target neurons. a** Schematic for ChR2-assisted functional mapping of NTS^TH → PVN connection using ex vivo electrophysiology. **b** Representative loose seal traces from PVN neurons during NTS^TH axon photostimulation (blue shaded region). **c** Pie chart summarizing percent of PVN neuron response types to NTS^TH photoactivation (n = 428 neurons, 6 mice). **d** Representative agarose gel of RT-PCR outcomes assessing MC4R expression from single PVN neurons collected following patch clamp recordings. **e** Summary of PVN MC4R+ and MC4R- neurons activated (n = 13 cells, 3 mice) or inhibited (n = 13 cells, 3 mice) by NTS^TH axon photo-stimulation. **f** Schematic for assessing NTS^DBH → PVN^MC4R connection in vivo. **g** Photomicrograph of Chrimson (red) expression in NTS^DBH soma and PVN projections, GCaMP8m (green) transduced PVN^MC4R neurons, and fiber tip (FT) (scale bars: 500 μm, 200 μm). **h–j** Effect of NTS^DBH → PVN projection photostimulation (10 s, 10 Hz) on PVN^MC4R GCaMP8m fluorescence in control (n = 3 mice) or Chrimson transduced (n = 6 mice) animals. **h** Average traces, **i** individual temporal heatmaps, and **j** z-score quantification during pre-stimulation, stimulation (red line), and post-stimulation periods (2-way RM-ANOVA with Tukey's multiple comparison, *p = 0.0141). **k–m** PVN^MC4R GCaMP8m fluorescence during train photostimulation of NTS^DBH → PVN axons before and after intraperitoneal yohimbine + prazosin cocktail administration (n = 5 mice). **k** Average traces, **l** individual temporal heatmaps, and **m** quantification during stimulation period (two-tailed, paired t-test, *p = 0.038). **n** Schematic for ex vivo cell-type specific Chrimson-assisted circuit mapping of NTS^DBH → PVN^MC4R connection. **o** Representative images showing expression of Chrimson in NTS^DBH soma and PVN axons, and GFP transduced PVN^MC4R (scale bars: 500 μm, 200 μm, 50 μm). **p** Representative loose seal traces and **q** summary of PVN^MC4R neurons activated or inhibited by NTS^DBH axon photostimulation (n = 73 neurons). Data are presented as mean values +/− SEM.

## PVN^MC4R neurons are targeted by ascending NTS^TH axons

The PVN harbors a diverse set of molecularly defined neuron ensembles involved in feeding, glucose counterregulation, and several other autonomic responses. We next sought to identify how NTS^TH input affects PVN neuronal activity. For this, we initially performed channelrhodopsin-2 assisted circuit mapping (CRACM)[29] with loose seal recordings followed by single-cell genotyping (Fig. 5a). In acute slices, photostimulating NTS^TH axons influenced PVN neuron firing heterogeneously, with many non-responsive neurons and others exhibiting increased or decreased firing rates (Fig. 5b, c). Using pharmacologic inhibition, we found that excitatory NTS^TH → PVN connection requires both glutamatergic and nor/adrenergic transmission, whereas inhibitory NTS^TH → PVN connection requires both α1 and α2 adrenergic receptors (Supplementary Fig. 7a–h).

To determine the molecular identity of PVN targets, we collected a subset of PVN neurons that were responsive to NTS^TH axon photo-stimulation. Based on the known role of PVN^MC4R neurons in appetite regulation[30], we gauged for the presence of *Mc4r* transcript. We found that NTS^TH form both excitatory and inhibitory connections with PVN^MC4R (Fig. 5d, e). To verify and further characterize a possible NTS → PVN^MC4R connection, we generated *Dbh-flp::Mc4r-cre* double transgenic mice. We next targeted flp-dependent Chrimson and cre-dependent GCaMP8m to the NTS and PVN, respectively, and placed an optical fiber over the PVN (Fig. 5f, g). Fiber photometry recording in awake mice showed that continuous 10 s stimulation of NTS^DBH axon terminals in the PVN evoked transient PVN^MC4R activation followed by prolonged inhibition (Fig. 5h–j). Consistent with ex vivo recordings, the inhibitory phase of this connection was sensitive to systemic α-adrenergic blockers (Fig. 5k–m).

We next validated this connection ex vivo using NTS^DBH → PVN^MC4R CRACM. For this, we delivered flp-dependent Chrimson to the NTS and cre-dependent GFP to the PVN (Fig. 5n, o). In agreement with blind electrophysiology recordings and in vivo photometry results, fluorescent targeted cell-attached recordings revealed that NTS^DBH heterogeneously effects PVN^MC4R activity (Fig. 5p, q). Consistent with recordings from randomly selected PVN neurons, we found that NTS^DBH → PVN^MC4R connections utilize nor/adrenergic mechanisms. Recordings with pharmacological blockers showed that excitatory

NTS^TH neurons. Intraperitoneal injection of 2DG dramatically increased circulating corticosterone, which was attenuated by inhibition of rostral, but not caudal, PVN projecting NTS^TH neurons (Fig. 4h). Inhibiting rostral-NTS^TH → PVN projection also tended to decrease 2DG-induced rise in blood glucose but did not reach statistical significance (Fig. 4i). Collectively, these results illustrate a key role for PVN projecting rostral NTS^TH neurons in feeding and hypoglycemic response.

NTS$^{DBH}$ → PVN$^{MC4R}$ connection requires glutamatergic and nor/adrenergic transmission, while inhibitory NTS$^{DBH}$ → PVN$^{MC4R}$ connection utilizes α-adrenergic signaling (Supplementary Fig. 7j–o). Collectively, our findings reveal heterogeneous NTS$^{DBH}$ → PVN$^{MC4R}$ connection dynamics in which nor/adrenergic signaling is required for prolonged inhibition.

## Nor/adrenergic modulation of ARC$^{AgRP}$→PVN$^{MC4R}$ connection

Given the sensitivity of NTS$^{DBH}$ → PVN$^{MC4R}$ connection to adrenergic blockers, we performed fluorescence-guided ex vivo patch clamp recordings from MC4R/tdTomato reporter mice to directly interrogate nor/adrenergic modulation of PVN$^{MC4R}$ neurons. Loose seal recordings with synaptic blockers revealed direct heterogenous NE regulation of PVN$^{MC4R}$ activity (Supplementary Fig. 8a, b). However, application of NE with intact synaptic signaling dramatically increased the magnitude of PVN$^{MC4R}$ neuronal response and the number of responsive neurons for both activation and inhibition (Supplementary Fig. 8c). These results suggest that while nor/adrenergic regulation of PVN$^{MC4R}$ neurons can occur directly, this effect and the number of responsive neurons is amplified by indirect modulation of presynaptic release (Supplementary Fig. 8d). Consistent with presynaptic modulation, bath application of NE potentiated inhibitory synaptic input onto PVN$^{MC4R}$ neurons with an overall increase in IPSC frequency in whole-cell voltage clamp recordings (Fig. 6a–c).

A major source of appetite-regulating inhibitory input onto PVN$^{MC4R}$ neurons arrive from hunger-sensitive ARC$^{AgRP}$ neurons. ARC$^{AgRP}$ neurons are responsive to NE[31]; however, whether their synaptic terminals are directly responsive to NE/E is unknown. Therefore, we next assessed the effect of NE on ARC$^{AgRP}$ → PVN connection with CRACM. For this, we selectively expressed ChR2 in ARC$^{AgRP}$ neurons of *Agrp-ires-cre* mice and recorded axonal light-evoked synaptic currents from PVN neurons (Fig. 6d). Bath application of NE robustly increased photoevoked eIPSC amplitude suggesting that ARC$^{AgRP}$ → PVN connection can be potentiated by presynaptic nor/adrenergic signaling (Fig. 6e, f). To verify whether NE has a similar effect in vivo, we expressed Axon-GCaMP6s in AgRP neurons and performed fiber photometry recording from ARC$^{AgRP}$ axon terminals in the PVN (Fig. 6g). Intraperitoneal administration of the α1-adrenergic receptor agonist, cirazoline, significantly and reversibly activated ARC$^{AgRP}$ axons in the PVN (Fig. 6h–j). Since PVN$^{MC4R}$ neurons are a major target of ARC$^{AgRP}$ neurons, α1-adrenergic receptor dependent activation of ARC$^{AgRP}$ axon terminals would be expected to suppress PVN$^{MC4R}$ neuron activity. Consistently, fiber photometry recording from *Mc4r-cre* mice revealed that cirazoline robustly inhibited PVN$^{MC4R}$ neuron activity (Fig. 6k–n). Taken together, these results suggest that medullary nor/adrenergic input gates presynaptic inhibitory tone onto PVN$^{MC4R}$ neurons arriving from ARC$^{AgRP}$, a key connection previously implicated in feeding behavior[30].

## PVN$^{MC4R}$ neuronal suppression is necessary for hypoglycemic feeding

Previously, PVN$^{MC4R}$ neurons were shown to be required for daily food intake regulation;[30] however, their role in hypoglycemic feeding is not known. Based on our results showing an orexigenic NTS$^{TH}$ → PVN connection (Fig. 3), the importance of NTS$^{TH}$ → PVN in glucoprivic feeding (Fig. 4), and identification of a NTS$^{DBH}$ → PVN$^{MC4R}$ circuit (Fig. 5), we examined the role of PVN$^{MC4R}$ in hypoglycemic feeding. We first monitored whether NE/E is released onto the PVN$^{MC4R}$ neurons during glucoprivation and found that 2DG injection caused a significant increase in GRAB$_{NE}$2h signal from these neurons (Fig. 7a–d). Consistent with an overall inhibitory impact of NE/E on these neurons, monitoring in vivo PVN$^{MC4R}$ activity with GCaMP7s revealed robust suppression by 2DG-induced glucoprivation (Fig. 7e–h). Next, we chemogenetically activated PVN$^{MC4R}$ to assess the necessity of their inhibition in hypoglycemic feeding. We found that simultaneous PVN$^{MC4R}$ activation

completely blocked food consumption caused by 2DG (Fig. 7i–k). These results show that PVN$^{MC4R}$ suppression is essential for hypoglycemic hunger.

## Discussion

It has been over six decades since Grossman has shown that NE injection into the hypothalamus elicits strong feeding behavior[32]. Subsequent studies established PVN as a major site for NE action in eliciting feeding, whose magnitude is on par with another well-known orexigenic agent, neuropeptide-Y (NPY)[33]. However, neither the source/target nor the physiological role of PVN NE/E has been fully understood. Here we show that PVN nor/adrenergic signaling is dynamically regulated by nutritional state. Using the GRAB$_{NE}$2h sensor selectively expressed in PVN neurons, we found that hunger elevates baseline extracellular NE/E levels and food access suppresses it after a short peak. This pattern is mirrored by activity of nor/adrenergic axons arriving from the rostral NTS to PVN. Selective optogenetic activation and chemogenetic inhibition of rostral NTS$^{TH}$ → PVN connection revealed contribution of this pathway to feeding after deprivation and in response to hypoglycemia. Voracious feeding in response to NTS$^{TH}$ → PVN activation also involved strong increase in motivation to obtain food, which was comparable to ARC$^{AgRP}$ stimulation. Consistently, we found ARC$^{AgRP}$ → PVN$^{MC4R}$ connection is potentiated by activation of presynaptic α1-adrenergic receptors on AgRP axon terminals. Collectively these findings suggest that information related to peripheral hunger signals are transmitted by catecholaminergic NTS neurons and integrated in the PVN melanocortin pathway, revealing that the NTS$^{TH}$ → PVN$^{MC4R}$ circuit partake in energy deficit response to food deprivation and hypoglycemia.

Consistent with earlier neuroanatomical studies, we found that NTS$^{TH}$, but not LC$^{TH}$, neurons are a major source for NE/E in PVN[34,35]. We have previously shown that ARC projecting collaterals from these neurons stimulate hypoglycemic feeding, whereas global activation of entire NTS$^{TH}$ neuron population was established to be strongly appetite suppressive[14,17,36]. The anorexigenic response elicited by these neurons appears to be mediated through projections to DMH and PBN[16,17]. Contrary to our findings, recent studies have shown that NTS$^{TH}$ projections to PVN are also mildly anorexigenic, at least in rat[18,37]. However, these studies were focused on A2 adrenergic population in the caudal NTS and likely excluded the rostral C2 group. Indeed, evidence suggests that TH neurons may have opposite effects on appetite depending on their precise location in the NTS[15] such that rostral NTS$^{TH}$ neurons (C2) being orexigenic and the caudally located ones (A2) are anorexigenic. Regardless, these studies demonstrate that NTS$^{TH}$ neurons are not a monolithic population and significant functional heterogeneity exists along the rostro-caudal axis.

Functional divergence among rostro-caudal NTS$^{TH}$ neurons and the complex response profile on its PVN targets might reflect multi-modal nature of this projection in responding to various stressors[26,38]. In addition to hypoglycemia, earlier cFos labeling studies showed that NTS$^{TH}$ neurons can be activated by a variety of physiological and psychological stressors. In line with this, several other PVN neuronal populations have been reported to encode multiple behavioral states[39]. Stress has a complex relation to appetite; while some acute stressors tend to be anorexigenic, others can drive overeating of rewarding food, which is thought to mitigate negative emotions associated with stress, eventually contributing to obesity[40–46]. Coping with stressful challenge can also be energy demanding and the orexigenic drive by stress activated pathways such as NTS$^{TH}$ → PVN may have adaptive benefits to restore consumed resources. Notably, appetite-promoting effect of rostral-NTS$^{TH}$ → PVN activation is sustained long after the stimulation ceases. The lingering hunger may help ensure replenishment of energy stores lost during the stressful experience especially if feeding was not optimal through the process. Further imaging and functional studies with increased resolution are needed to

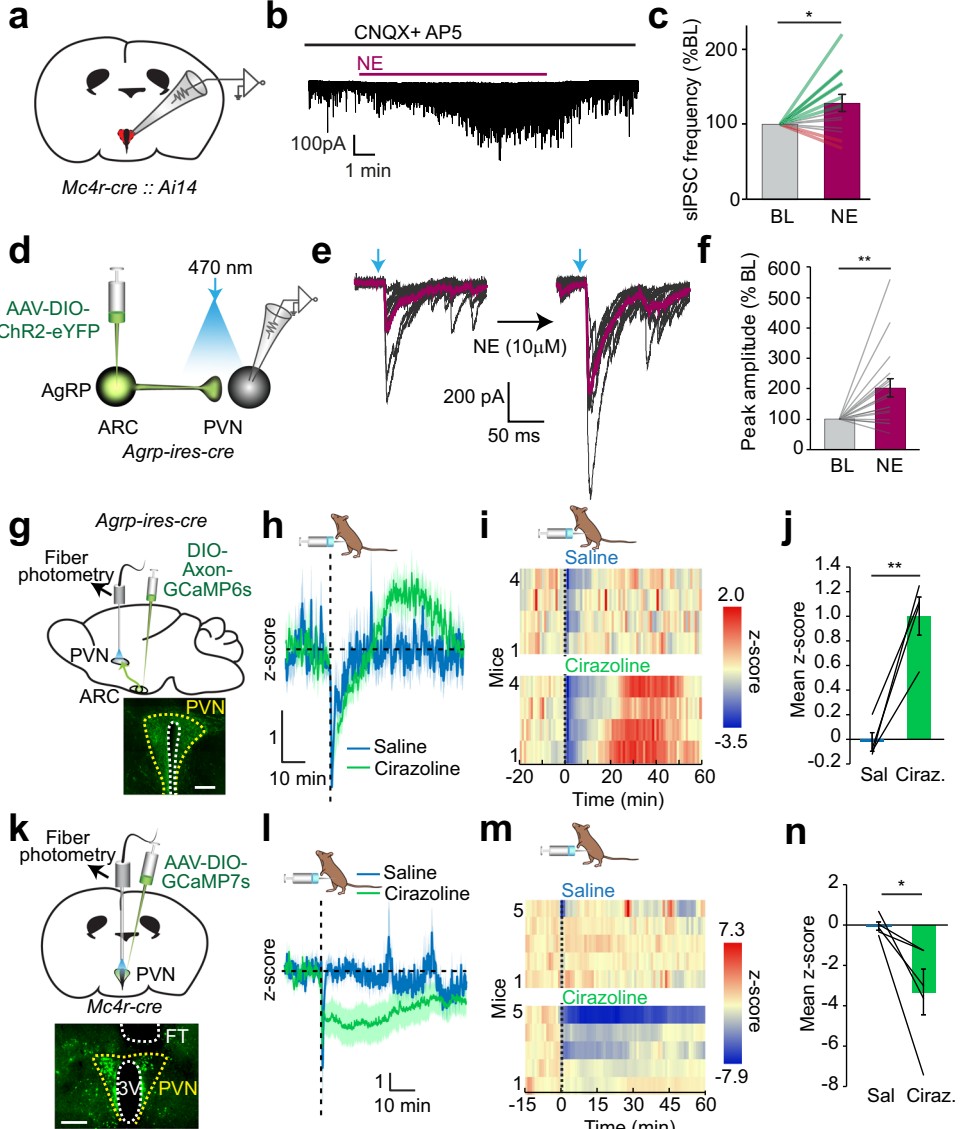

**Fig. 6 | Nor/adrenergic modulation of inhibitory ARC^AgRP → PVN connection at the presynaptic terminal. a** Schematic for fluorescence-guided patch clamp recording from genetically tdTomato labeled PVN^MC4R neurons. **b** Representative sIPSC trace from a PVN^MC4R neuron treated with NE. **c** Relative change in sIPSC frequency during baseline (BL) and after NE application ($n = 14$ cells, 5 mice, two-tailed paired t-test, *$p = 0.028$). **d** Schematic for electrophysiological assessment of ARC^AgRP → PVN eIPSC connection strength with ChR2-eYFP. **e** Individual (black) and average (purple) light-evoked ARC^AgRP → PVN eIPSC before and after NE bath application. **f** Relative amplitude of light-evoked ARC^AgRP → PVN eIPSC after NE application ($n = 16$ neurons, 6 mice, two-tailed, paired t-test, **$p = 0.0036$). **g** Schematic for recording in vivo ARC^AgRP → PVN projection activity with Axon-

GCaMP6s and representative image of Axon-GCaMP6s expressing ARC^AgRP axons in the PVN (scale bar: 200 μm). **h–j** Effect of intraperitoneal cirazoline (0.5 mg/kg) administration on ARC^AgRP → PVN axonal Ca^2+ content ($n = 4$ mice). **h** Average traces, **i** temporal heatmaps from individual mice, and **j** quantification of mean z-score value from 20–50 min (two-tailed paired t-test, **$p = 0.0074$). **k** Schematic for in vivo GCaMP7s monitoring of PVN^MC4R activity and representative image of GCaMP7s expression (scale bar: 250 μm). **l–n** Effect of intraperitoneal cirazoline administration on PVN^MC4R GCaMP7s fluorescence ($n = 5$ mice). **l** Average traces, **m** individual temporal heatmaps, and **n** quantification of mean z-score values (two-tailed paired t-test, *$p = 0.040$). Data are presented as mean values +/− SEM. For specific p-values, data and statistics, please see associated Source data file.

establish whether the same subsets of NTS^TH neurons are recruited by various homeostatic challenges, including hunger, and how this information is conveyed to PVN neurons.

Ablation of hypothalamic nor/adrenergic fibers have long been known to disrupt feeding in response to a specific type of stress: hypoglycemia. While most existing research focuses on the role of VLM[47–49], our results suggest that orexigenic response to hypoglycemia strongly recruits dorsal medullary catecholamine neurons as well. Several lines of evidence support involvement of these neurons in hypoglycemic feeding: (1) rapid activation of NTS^TH projections to PVN by 2DG administration, (2) optogenetic activation NTS^TH → PVN is sufficient to elicit robust feeding response and corticosterone

release, (3) chemogenetic inhibition reduces, but does not eliminate, deprivation and 2DG-induced feeding. Collectively these results show that rostral NTS^TH neurons are critical for deprivation and counterregulatory responses. It remains to be addressed how the interplay between VLM and NTS catecholamine neurons coordinates hypoglycemic feeding. One possibility is that these parallel and partially redundant circuits might be recruited in response to different levels of hypoglycemic challenge and may have additive effect on counterregulatory feeding. Alternatively, the speed by which these pathways are recruited might differ or different aspects of counterregulatory response are activated by two cell groups. For example, a recent study showed counterregulatory glucagon release

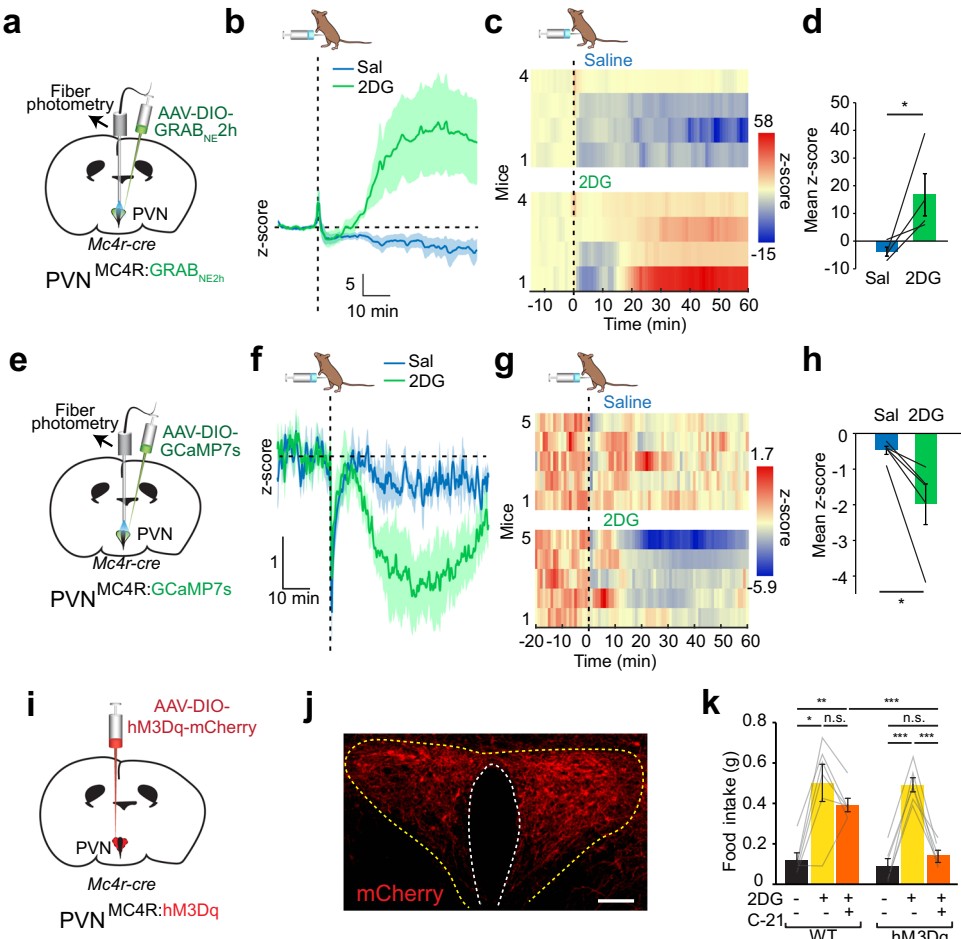

**Fig. 7 | PVN^MC4R inhibition is required for conterregulatory eating. a** Schematic of NE/E measurements from PVN^MC4R neurons. **b, c** Average fiber photometry trace (**b**) and individual heat maps (**c**) showing GRAB_NE2h signal in response to 2DG injection. **d** Summary graph for quantification of GRAB_NE2h fluorescence in PVN^MC4R neurons ($n = 4$ mice, two-tailed, paired t-test, *$p = 0.042$). **e** Schematic for monitoring PVN^MC4R activity in vivo with GCaMP7s. **f–h** Effect of intraperitoneal 2DG administration on PVN^MC4R GCaMP7s fluorescence ($n = 5$ mice). **f** Average traces, **g** individual temporal heatmaps, and **h** quantification of mean z-score from 10–60 min (two-tailed, paired t-test, *$p = 0.032$). **i** Schematic for chemogenetic PVN^MC4R activation with hM3Dq. **j** Representative image of hM3Dq expression in PVN^MC4R neurons (scale bar: 50 μm). **k** Effect of PVN^MC4R activation on 2DG-induced feeding in control ($n = 6$ mice) and hM3Dq expressing ($n = 6$ mice) animals (2-way RM ANOVA with Tukey's multiple comparison, ***$p < 0.009$, n.s.: Not significant). Data are presented as mean values +/− SEM.

to be mediated by ascending VLM input to SON^AVP neurons[50]. Glucose clamp studies can be combined with the axonal fiber photometry recording approach described here to establish how these diverse medullary populations partake in various aspects of counterregulation[51].

Notably, PVN NE/E signaling appears to contribute not only to hypoglycemic feeding but also to daily homeostatic hunger. Consistent with earlier microdialysis studies[10,11], we found that PVN NE/E levels drop with refeeding after a transient peak, which might be related to arousal. This suggests that prolonged hunger may elevate baseline levels of NE/E in PVN. Indeed, chronic GRAB_NE2h recording experiments showed slow rise in the ambient NE/E levels in the PVN in the absence of food. Additionally, in line with earlier observations in the ARC[52], ghrelin injection caused significant increase in NE/E levels in the PVN. Finally, we found that activation of AgRP neurons caused small but significant increase in PVN NE/E level. Taken together with the observation that NE potentiates GABA release from AgRP terminals, this surprising result suggests presence of a positive feedback loop between AgRP and PVN-projecting NTS^TH neurons, activated during hunger.

Our findings suggest that NTS^TH neurons are likely to be the source of elevated NE/E in PVN in response to deprivation-related signals. This is also in agreement with the presence of an NTS^TH

neuronal subset activated by ghrelin[14,53]. Finally, acute chemogenetic suppression of PVN projecting NTS^TH neurons reduced feeding. Interestingly, severing ventral noradrenergic bundle does not reduce daily food intake; on the contrary, these rats become hyperphagic and obese[54]. This suggests that chronic loss of orexigenic drive from this pathway, but not the intermingled anorexigenic fibers from A2, can be compensated.

Finally, activation of NTS^TH → PVN pathway by metabolic cues may not be limited to visceral signals. Given the established role of rostral-NTS in integrating gustatory signals, the initial activation NTS^TH → PVN projection might be mediated by cephalic phase signals, which might also be contributed by arousal signals. Indeed, when fasted mice were given inaccessible food, this initial rise in NE/E levels persisted, supporting the possibility of sensory origin. Given its strong orexigenic potential, the subsequent drop in NE/E signal is likely driven by reduced tonic activity in rostral NTS^TH input as feeding progresses. Since our recordings focused on the first ~40 min after food access, we may have overlooked satiation-dependent activation of caudal NTS^TH neurons, which are implicated in appetite suppression. Further research is warranted to explore how rostral versus caudal NTS^TH neurons participate in different phases of feeding.

Although adrenergic blockers abolished NTS^TH induced inhibition of PVN neurons, other transmitters might also be involved with

**Table 1 | AAV constructs used in this study**

| Company | Catalog # | Virus name |
|---|---|---|
| WZ Biosciences | YL003012-AV9 | AAV9-hsyn-DIO-NE2h |
| WZ Biosciences | YL003011-AV9 | AAV9-hsyn-NE2h |
| Addgene | 28306 | pAAV-FLEX-tdTomato (AAV1) |
| Addgene | 112010 | hSynapsin1-FLEX-axon-GCaMP6s (AAV5) |
| Addgene | 20298 | pAAV-EF1a-double floxed-hChR2(H134R)-EYFP-WPRE-HGHpA (AAV5) |
| Addgene | 154867 | pAAV-hSyn-fDIO-hM4D(Gi)-mCherry-WPREpA (AAV8) |
| Addgene | 87306 | pEF1a-DIO-FLPo-WPRE-hGHpA (Retrograde) |
| VVF Zurich | v413-9 | ssAAV-9/2-hSyn1-chl-dFRT-ChrimsonR-tdTomato(rev)-dFRT-WPRE-hGHp(A) |
| VVF Zurich | v663-8 | ssAAV-8/2-shortCAG-dlox-jGCaMP8m(rev)-dlox-WPRE-SV40p(A) |
| Addgene | 104495 | pGP-AAV-CAG-FLEX-jGCaMP7s-WPRE (AAV1) |
| Addgene | 44361 | pAAV-hSyn-DIO-hM3D(Gq)-mCherry (AAV2) |
| Addgene | 51502 | AAV pCAG-FLEX-EGFP-WPRE (AAV2) |

different activity patterns imposed on NTS[TH] axons. Indeed, Marie et al.[55] reported normal feeding in *dbh* deficient mice in response to low dose of insulin whereas at higher doses feeding was impaired, suggesting that lack of NE/E can be partially compensated by other signaling molecules during mild hypoglycemic challenges. One such candidate might be NPY, which is coexpressed by PVN projecting rostral NTS[TH] neurons. Interestingly, while global NPY knock-out mice show nearly complete loss of hypoglycemic feeding[56], selective ablation of NPY from DBH neurons only partially reduced feeding in response to activation of rostral NTS[DBH] neurons[15]. Considering the absolute requirement of DBH neurons for hypoglycemic feeding[22], these findings indicate that DBH neurons may also recruit NPY signaling from non-DBH neurons. This is consistent with our observation that ARC[AgRP] synaptic terminals, which also contain NPY[57], can be activated by nor/adrenergic signaling. Further support for a role for NPY comes from the long lasting orexigenic effect of NTS[TH] → PVN stimulation (Fig. 3o), which was previously shown to be mediated by this molecule[28].

A limitation of our study is the use of *cre/flp* driver lines line to access catecholamine neurons. Due to long post-surgery wait time, we cannot rule out the possibility that a small subset of labeled neurons might actually be TH negative in our experiments. However, based on recording and pharmacology experiments, which confirmed requirement of adrenergic transmission, we think that such misexpression is likely to be functionally inconsequential. We also did not directly test whether the NE/E release by NTS[TH] axons in the PVN increases feeding and inhibits PVN[MC4R] neurons. Future experiments with NTS[TH] specific inactivation of NE/E synthesis will help clarify this.

In summary, we identified a circuit organization activated by hunger and hypoglycemia. Thus far, nearly all tested dorsal medullary neuronal subpopulations reduced feeding[17,18,58–66]. Our results suggest that rostral NTS[TH] neurons are unique among the neighboring populations and associated with feeding response thereby providing evidence that this region is also a route for orexigenic signals. NTS[TH] neurons elicited a heterogenous activity pattern in PVN[MC4R] neurons, with dominant response being inhibition through, in part, facilitation of GABA release from ARC[AgRP] presynaptic terminals in PVN. Effectively, this organization connects vagal sensory input to the melanocortin pathway through NTS catecholamine neurons and plays an essential role in hypoglycemic feeding. Given its broad activation pattern in response to variety of stressors besides hypoglycemia[26,38], NTS[TH] axons may recruit additional PVN neuronal subpopulations, other than PVN[MC4R] and the previously described PVN[CRH] neurons, depending on the physiological challenge. Thus, it will be important to understand how the type and severity of different stressors are encoded in the NTS[TH] → PVN interoceptive-preautonomic axis with cell-type specific resolution using the strategies described here.

## Methods

### Animals
Experimental mice were housed on 12-h light and dark cycle at 20–24 °C, and 40–60% humidity, with ad libitum access to standard chow food and water, unless stated otherwise. Recombinase-expressing lines *Sim1-cre* (Jackson Labs Stock 006395), *Th-cre* (Jackson Labs Stock 008601), *Th-ires-cre*[67], *Agrp-ires-cre* (Jackson Labs Stock 012899), *ai14* (Jackson Labs Stock 007914), *Mc4r-cre* (Jackson Labs Stock 030759) and *Dbh-flp* (Jackson Labs Stock 033952) were back-crossed with C57BL/6 (Jackson Labs Stock 000664) for maintenance. Studies were performed with 6–24-week-old male and female mice, using similar numbers for both sexes in each experiment. Animal care and experimental procedures were approved by University of Iowa Institutional Animal Care and Use Committee (IACUC) and Istanbul Medipol University Animal Care Committee, MEDITAM. Mice welfare and health checks were conducted in accordance with the IACUC guidelines. Sentinel mice cages were periodically screened for pathogens. Mice that displayed unhealthy posture or more than 20% weight loss were removed from the study.

### Stereotaxic surgeries
**rAAV injections.** Stereotaxic surgeries were performed on 6–24-week-old male and female mice. Briefly, mice were anesthetized with 1.5% isoflurane in the stereotaxic instrument (David Kopf instruments, Tujunga-CA). Disinfected scalp was incised to expose skull which was then drilled to obtain a small hole for injection of 150 to 600 nL virus to each side intracranially using a pulled glass pipette (Drummond Scientific, Wiretrol, Broomall-PA) with -50 μm tip diameter. Viral injections were performed in the NTS (bregma: −7.00 mm, midline: ±0.50 mm, dorsal surface −3.50 mm), rostral NTS (bregma: −6.50 mm, midline: ±0.50 mm, dorsal surface −3.70 mm), caudal NTS (bregma: −8.20 mm, midline: 0 mm, dorsal surface −3.40 mm) and PVN (bregma: −0.75 mm, midline: ±0.25 mm, dorsal surface: −5.00 mm), as 50 nL/min by a micromanipulator (Narishige, East Meadow, NY), allowing 10 min spread time for each injection. Scalp was stitched after removing the pipette and placing the optical fiber. At least 4 to 8 weeks were given for animal recovery and transgene expression before further experiments. Viruses used are listed in Table 1.

**Optical fiber placement.** Ferrule capped optical fiber (200 μm core diameter, NA = 0.50, ThorLabs, for in vivo optical stimulation; 400 μm core diameter, NA = 0.48, Thorlabs, for fiber photometry) was implanted above the PVN (bregma: −0.75 mm, midline: 0 mm, dorsal surface: −4.8 mm) following viral injections and fixed with dental cement.

### In vivo fiber photometry recording and analysis
Free behaving mice were tethered through their ferrule implants to the fiber optic cables (400 μm core, 0.48 NA, bundled fibers, Doric Lenses)

using ceramic mating sleeves (Thorlabs) covered with black tubing to prevent light interference. Mice were allowed a day to acclimate to the tethers before the fiber photometry experiments and were tethered at least 30 min before recording. Fiber photometry signal was recorded at 3 Hz sampling rate, using Doric FP Bundle Imager (Doric Lenses). Light intensity for each wavelength at the end of the tip was set to be approximately 30–50 μW. We recorded a baseline of 10–30 min before any interference (i.e., intraperitoneal injection, food presentation etc.) and recorded for at least 30 min after the events. For chronic recording, data was acquired as 5 min at every 30 min. At the end of the experiments, mice were processed for post hoc histological evaluation. We eliminated animals where the viral transgene expression or the fiber tip was off target.

For the analysis, isosbestic signal (405 nm) was fit to the calcium dependent (465 nm) signal using the linear least squares fit using a custom MATLAB script. Then, ΔF/F values were calculated as ($F_{465 nm}$ − fitted $F_{405nm}$)/(fitted $F_{405nm}$). To account for the inter-animal differences in signal intensities, z-scores were calculated from these values as $(F - \mu_{(baseline)})/\delta_{(baseline)}$, where μ and δ are the average and standard deviation of the baseline period, respectively, while F is the signal for the given time.

For in vivo fiber photometry paired photostimulation experiments, data was acquired at 10 Hz, and photostimulation was performed at 10 Hz with 30 ms pulse width, for 10 s, after 2 mins of baseline period using red laser (635 nm, Doric Lenses Inc., Quebec), and at least five times for each animal. For stimulation following antagonist administration, animals were intraperitoneally injected with saline as control or a cocktail of 2 mg/kg yohimbine and 1 mg/kg prazosin 10 min before the first record. After 2 min of baseline recording, photostimulation was performed as 10 x (3 s ON and 5 s OFF, 10 Hz, 30 ms pulse width) and the stim protocol was repeated as a total of 3 times, having 3 mins of recovery time in between. Analysis of photostimulation paired fiber photometry data was performed same as explained with the exception that fitted 405 nm was replaced with baseline average for the ΔF/F calculation.

## Behavioral experiments

**Food intake.** Mice were placed in custom made plexiglass cages with free access to food and water, unless being fasted and were habituated to handling and/or being tethered to optical fibers for 2–3 days. Photostimulation was performed using a 473 nm diode laser (Doric Lenses Inc., Quebec) with 1 s stimulation (10 Hz, 10 ms pulse width) repeated every 4 s, for a total of 2–4 h. Food intake was measured manually every 2 h for baseline, stimulation and post-stimulation periods, or continuously using FED3 pellet dispensers[68]. For inhibition experiments, mice were fasted for 18 h, and then were injected with saline or CNO (3 mg/kg) and food intake was manually measured for 4 h. In another set of experiments, food intake was also measured after intraperitoneal injection of 2DG (450 mg/kg) with or without CNO (3 mg/kg)/DCZ (0.5 μg/kg) and food intake was measured for up to 6 h manually, or with FED3 pellet dispensers.

For all experiments, daytime measurements were started within 2 h of light onset, while nighttime measurements were started immediately with dark onset, baseline being the dark onset on the previous night. During a subset of the experiments, mice were traced with a CCD camera and the videos were analyzed using Ethovision XT15 software for the time spent in, latency to enter and number of entries to the food area.

For feeding pattern analysis, food intake was monitored continuously using Coulbourn Habitest or FED3 single pellet food delivery systems. A "meal" was defined as 4 or more pellets consumed within 10 min. Mice treated with mifepristone were given an i.p. injection of the drug (5 mg/kg) freshly dissolved in DMSO, before photostimulation.

**Progressive ratio task.** After mice were acclimated to the use of FED3 devices, the FED3 setting was changed to 'Fixed Ratio-1' (FR1) mode, where mice had to poke their noses to one of the poker holes in the device to obtain food, then to FR3 (3 pokes per pellet) and finally to FR5 (5 pokes per pellet), changing the setting each day. Next day the device mode was set to progressive ratio schedule, that delivered chow pellets with a nose poke ratio of 1, 2, 4, 6, 9, 12, 15, 20, 25, 32, etc. (adapted from ref. 69), under ad libitum condition with or without photostimulation, or after overnight fasting. The break point was defined as the number of pellets where animals stopped working for more than 10 min.

## Blood glucose and corticosterone measurements

Blood was drawn following a tail snip and either immediately evaluated for blood glucose with CareTouch Blood Glucose Monitoring System or were processed for corticosterone measurement with Alpco Mouse and Rat Corticosterone ELISA kit, as described in the user manual. Blood was drawn 10 min before and 60 min after photostimulation, or 10 min before and 90 min after 2DG and/or DCZ injections.

## Electrophysiology

2- to 4-month-old male and female mice were used for ex vivo electrophysiology. Mice deeply anesthetized with isoflurane were decapitated and 300-μm-thick coronal sections were obtained from a vibratome in a chilled 95% $O_2$ and 5% $CO_2$ aerated cutting solution consisting of (in mM): 92 NMDG, 92 HCl, 2.5 KCl, 1.2 $NaH_2PO_4$, 30 $NaHCO_3$, 20 HEPES, 25 glucose, 5 sodium ascorbate, 2 thiourea, 3 sodium pyruvate, 10 $MgSO_4$, and 0.5 $CaCl_2$. Brain sections were incubated for 5 min in the same aerated solution warmed at 33 °C. In the same heated solution, NaCl was added (NaCl concentration in solution 20 μM) and slices were incubated for 5 more minutes. More NaCl was added (final NaCl concentration 50 μM) and sections were incubated an additional 2 min. Brain slices were transferred and incubated for >45 min at room temperature in an aerated solution containing (in mM): 92 NaCl, 2.5 KCl, 1.2 $NaH_2PO_4$, 30 $NaHCO_3$, 20 HEPES, 25 glucose, 5 sodium ascorbate, 2 thiourea, 3 sodium pyruvate, 2 $MgSO_4$, and 2 $CaCl_2$. Slices were placed in a recording chamber perfused with an aerated artificial cerebral spinal fluid solution (aCSF) consisting of (in mM): 124 NaCl, 2.5 KCl, 1.2 $NaH_2PO_4$, 24 $NaHCO_3$, 5 HEPES, 12.5 glucose, 2 $MgSO_4$, and 2 $CaCl_2$. Cell-attached and whole-cell patch clamp recordings were performed using electrodes with 5–10 MΩ tip resistances. Pipettes were filled with the same aCSF solution for cell-attached recordings. For whole-cell voltage clamp recordings the internal solution was composed of (in mM): 125 CsCl, 5 NaCl, 10 HEPES, 0.6 EGTA, 10 lidocaine ethyl bromide, 5 Mg-ATP, and 0.4 $Na_2$-GTP. Neurons were randomly selected in the PVN for recordings. PVN neurons expressing GFP or tdTomato were selected for fluorescence guided recordings. Data was acquired with MultiClamp 700B Amplifier (Molecular Devices, San Jose, CA) and analyzed with Axon pCLAMP 11 software (Molecular Devices, San Jose, CA). To activate ChR2 and Chrimson opsins in axons, 470 nm laser stimulation was used with an LED array (Doric Lenses Inc., Quebec) and laser pulses were generated with an AMPI Master-9 Pulse stimulator. For ex vivo optogenetic experiments, a 10 Hz/3 ms pulse width photostimulation protocol was used.

PVN neuron responses to different stimuli were classified as excitatory or inhibitory if firing rate increased or decreases by at least 20% from baseline, respectively. GABAergic signaling was blocked with picrotoxin (10 μM). Glutamatergic signaling was blocked with CNQX (10 μM) and D-AP5 (40 μM). Alpha-adrenergic receptors were blocked with prazosin (2 μM) and yohimbine (2 μM). To determine the effects of nor/adrenergic signaling on PVN neuron activity, 5 min of baseline was recorded before application of NE (10 μM) followed by 10–15 min of continued recording. To identify PVN cell-types influenced by $NTS^{TH}$ axon photostimulation, a subset of cells with consistent responses

were collected into the pipette tip by applying suction and processed for single-cell PCR. For ex vivo assessment of NTS$^{DBH}$ → PVN$^{MC4R}$ connection, cell-attached recordings were performed. For ChR2- and Chrimson-assisted circuit mapping, 3–5 sweeps were obtained and neurons with consistent response types were characterized as activated or inhibited. For frequency analysis to test blocker sensitivity, entire 0–10 s (excitatory) and 0–3 or 0–10 s (inhibitory) stimulation period is used to calculate firing rate. Whole-cell IPSC recordings were performed with voltage clamped at −60 mV and the average response to 5 stimulations was calculated for each cell during baseline period and after NE application.

## Single-cell rt-PCR

Following electrophysiological recordings, the contents of the cells were aspirated into patch clamp pipettes and the pipette contents were immediately transferred to microtubes containing RNA extraction buffer (PicoPure RNA Extraction Kit, Thermo Fisher Scientific). Total RNA was obtained as described in the user's manual. First strand cDNA synthesis (RevertAid First Strand cDNA Synthesis Kit, Thermo Fisher Scientific) was followed by PCR in 20 µl reaction buffers, containing 1 µl of the single-cell cDNA and the specific primers for *Mc4r* mRNA (Forward: 5′-CGCGCTCCAGTACCATAACA-3′, Reverse: 5′-ATGGTCAAGGTAATCGCCCC-3′) and *Gapdh* mRNA (Forward:5′-GTCGGTGTGAACGGATTT-3′, Reverse:5′-ATGTTAGTGGGGTCTCGCTC-3′) were used for amplification in separate reactions. After two rounds of PCR (30 cycles each), products were monitored with 2% agarose gel.

## Post hoc analysis

Mice were anesthetized with 1.5% isoflurane and were transcardially perfused with 4% paraformaldehyde (PFA) and decapitated. Extracted brains were further fixed in 4% PFA for 4 h and were preserved in 30% sucrose solution. 75 µm brain sections were collected with a vibratome and were either directly mounted (Fluoroshield™ with DAPI, Sigma) or further processed for staining using anti-TH (1:5000, Abcam, #ab112), anti-cFos (1:5000, Cell signaling, #2250) or anti-mCherry (1:1000, Abcam, #ab167453) primary antibodies, and goat anti-rabbit IgG (H + L) Alexa Flour 488/568 (1:500, Thermo-Fisher) secondary antibodies. Imaging was performed with Olympus slide scanner.

## Statistical analysis and reproducibility

The number of mice in each cohort were estimated based on pilot experiments and previously published work[14,70]. Data distribution was assumed to be normal; however, this was not formally tested. Outlier tests were not performed. Where possible, experiments were repeated with an independent cohort, and results were combined for final analysis. Mice were randomly assigned to groups. Initial planning of experimental groups was not blinded, but where possible, investigators were blinded for the experiment and data analysis. Differences between two groups were tested with two-tailed paired and unpaired Student's t-tests. Multiple group statistical comparisons were made by one-way ANOVA or two-way repeated measures ANOVA, and corrected *p*-values, and two-tailed Pearson r and *p*-values were calculated using GraphPad Prism 9 (GraphPad Software Inc.). *n* represents mice or neuron numbers as indicated for each experiment. The representative images and recording traces provided in the figures were similar for the *n* number of repetitions provided in relevant sections. A *p*-value < 0.05 was considered to be statistically significant.

## Reporting summary

Further information on research design is available in the Nature Portfolio Reporting Summary linked to this article.

## Data availability

The data generated in this study are provided in the Source data file. Source data are provided with this paper.

## Code availability

Custom MATLAB scripts used for analysis of data is available upon request.

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

## Acknowledgements

This work is supported by NIH to D.A. R01DK126740.

## Author contributions

N.S.A. and C.L. performed fiber photometry recording; N.S.A., C.L., and I.A. performed behavioral experiments; N.S.A., I.C., C.L., and I.A. performed surgeries; N.S.A. and C.L. performed data analysis; N.S.A. performed genotyping and RT-PCR and prepared MATLAB code for data acquisition and analysis; D.D. managed mouse handling; C.L., Y.Y., and T.A. performed electrophysiological recordings; H.K., J.R., N.S.A., and F.K.A. performed imaging; H.K., I.A., J.R., N.S.A., C.L., U.S., H.C., M.I.A., and B.Y. contributed to post hoc analysis and provided reagents; N.S.A., prepared figures; D.A., N.S.A., and C.L. conceived experiments; D.A., N.S.A., and C.L. wrote the manuscript.

## Competing interests

The authors declare no competing interests.
