## [Peer Review File · Nature Communications]

Adrenergic Modulation of Melanocortin Pathway by Hunger SignalsREVIEWER COMMENTS

Reviewer #1 (Remarks to the Author):

Sayar-Atasoy and colleagues report results from a series of experiments designed to target nor/adrenergic NTS neurons that project to the PVN, and investigate their potential roles in feeding behavior after fasting and a variety of cell-specific experimental manipulations. Their reported data are generally consistent with other reports indicating that nor/adrenergic inputs to the PVN are functionally and anatomically heterogeneous, and participate in physiological and behavioral responses that underlie the central control of energy balance. Novel data reported here further implicate MC4R-expressing PVN neurons in responses to nor/adrenergic inputs that drive food intake.

The authors interpret their experimental results in a manner that is largely appropriate, and consistent with their data. These results should be of interest to a wide audience of researchers working on food intake and energy balance control. However, the importance of these findings could be strengthened by addressing or clarifying several points, as indicated below:

1. The first experiment includes a sophisticated in vivo assessment of extracellular NE/E within the PVN, achieved by measuring fluorescent signaling responses in Sim1 PVN neurons expressing AAV-DIO-GRABNE2h sensor.
 - a. The source of AAV-DIO-GRABNE2h for these experiments should be indicated.
 - b. Line 99-101: what was the inedible control object, and was it novel? (same question for experiment illustrated in Fig. 2D-G). Transient NE/E signaling (and Ca²⁺ signaling) in response to different stimuli is likely modulated by physiological state (i.e., fed vs. fasted) in a manner that impacts stimulus salience during each state.
 - c. In Fig. 1D, both the control object and food produced initial spikes in NE/E sensing by transfected Sim1 neurons, although the increase was maintained for a few more minutes before falling back towards baseline during P1 in mice that (presumably) continued to interact with and consume the food while reversing their state of deprivation. NE/E signaling then fell below the pre-P1 baseline during P2, i.e., after 10 min of food access. One might interpret the apparent decline in extracellular NE/E during P2 as a return to a lower baseline of NE/E signaling under non-fasted conditions, as discussed (lines 358-361). To support or challenge this interpretation, baseline signaling in the same mice under fasted vs. non-fasted conditions could be compared, assuming that the same mice were used for all conditions (Fig. 1; although whether the same mice were used is not explicitly stated). Similar comments apply for Ca²⁺ signaling in experiments producing data shown in Fig. 2. The use of Z-scores for these NE/E and Ca²⁺ measures is useful to normalize different baselines across animals, but within-subjects effects could/should be quantified. Such data could be used to support the statement (lines 176-177; 296-297) that fasting promotes NE/E signaling to PVN neurons.

d. Lines 102-103: ghrelin and insulin are described as "rapidly" increasing PVN NE/E levels, although the data in Fig. 1 indicate that the effects occur on different and rather slow timescales. Potential bases for these different timescales should be discussed. The general significance of these observations is not clear, given previous reports that systemic administration of CCK-8 (a known satiety hormone) and various stressful stimuli (including 2-DG) are known to increase extracellular levels of NE in the PVN, at least in rats. Are all of these NE/E responses to different treatments attributable to the recruitment of hindbrain NE/E neurons by "homostatic challenges" (i.e., stress)? This is discussed indirectly in lines 332-339; more explicit links could be developed.

e. As an overall comment regarding interpretation of results of stimulating NE/E signaling: Perhaps stimulation promotes PVN-mediated physiological outcomes (e.g., sympathetic or HPA axis activation), which then leads to increased feeding motivation via distributed neural circuits?

2. The terms noradrenergic and nor/adrenergic are used interchangeably throughout the manuscript. The applied models and methods (e.g., TH immunolabeling, TH-Cre mice, GRAB-NE sensing) do not discriminate noradrenergic (i.e., A2) from adrenergic (i.e., C2/C3) neurons within the NTS or their axon terminals within the PVN. For this reason, nor/adrenergic seems to be the preferred terminology.

3. With regards to the comment above (#5), published work indicates that the TH-Cre and TH-ires-Cre mice used in this study display Cre expression by neurons that do not express TH in adult mice (although they may have earlier in development). This also is evident in Fig. 2B (and S1A), although the authors describe the data as evidence for tdTom expression "specifically in catecholaminergic neuron soma" (line 122). What proportion of fluorescent reporter-labeled NTS cells are TH+? The authors should report these data and acknowledge that dorsal hindbrain neurons in addition to NTS nor/adrenergic neurons were likely transfected in experiments that depend on Cre expression. This does not invalidate the authors' experimental strategies or results, but interpretation of results from all experiments that depend on phenotypic specificity of Cre (and FLP) expression should be more conservative, taking the limitations of the models into consideration. For example, it would be relevant to know what proportion of mCherry+ neurons were TH+ in the experimental results shown in Fig. 4A, E; in these panels, mCherry reporter labeling seems more extensive than one might expect given the intersectional viral strategy.

4. Fig. 2I shows a consistent fall in baseline Ca²⁺ signaling from NTS axons within the PVN during the first 10min after 2DG injection. Conversely, Fig. 1H depicts less reduction of NE/E signaling below baseline within the PVN during the same timeframe after 2DG (except for a rapid decline in one mouse). The bases for this apparent timing difference in Ca²⁺ and NE/E signaling should be discussed.

5. Fig. 3A – what is the red non-nuclear labeling visible in the enlarged inset? What proportion of eYFP+ NTS neurons are TH+ after mid-NTS injection, and after anterior NTS injection? (related to point 3, above).

6. It's not clear why control mice took 100 min to enter the feeding area (Fig. 3H), or why mice with optogenetic stimulation still took ~20 min to enter the feeding area. Were all mice pre-acclimated/familiarized with the pellet delivery system? Why the slow timescale of approach and feeding? (even in non-fasted mice, one might expect a rapid inspection of the feeding area, rather than what looks like avoidance of it, Fig. 3G, H, I).

7. Fig. 3E is presented as if cell-attached recordings of NTS neurons were performed during PVN optogenetic stimulation and simultaneous food intake in the same mice, although the described ephys is performed *ex vivo*.

8. The progressive ratio experiment (data in Fig. 3Q-S) is difficult to understand. Fasted mice (red) apparently earned 10 pellets with 10 nose pokes, on average, while mice with optogenetic stimulation of rNTS TH+ neurons earned 10 pellets with about 7 pokes, on average. This doesn't fit with the described progressive ratio strategy (lines 691-695).

9. Figure 4 (and related Fig. S3, in which the distribution of fluorescent reporter-labeled neurons seems quite non-specific to TH+ neurons): While the overall data support differential effects of rostral vs. caudal NTS targeting, this interpretation could be strengthened by documenting the phenotypic specificity of flp-dependent NTS neuronal transfection. The effect sizes in B and C are rather small, with the experimental effect in B seemingly due to a higher baseline intake in the saline-injected control condition in hM4Di mice. In addition, the hM4Di mice seem more responsive to the hypophagic and corticosterone effects of 2DG compared to the eYFP control mice. The magnitude of feeding suppression after CNO administration seems similar between the eYFP control and hM4Di-expressing mice, and the effect of DCZ to attenuate the cort response in mice with anterior NTS hM4Di expression seems largely due to the greater cort response to 2DG in these mice (H). Effect sizes for these within-subjects data should be reported, and then compared between groups. Finally, why were different fluorescent reporter control viruses used (eYFP as control virus, but mCherry expressed with hM4Di)?

10. Fig. S1, the location of LC is incorrectly schematized

11. Fig. S3, why is mCherry labeling depicted in green channel?

Reviewer #2 (Remarks to the Author):

Following up on their previous observations that some Th-expressing NE neurons in the NTS promote food intake in response to fasting and glucoprivation and that projections to the ARC may explain some, but not all of this effect, the authors have now examined roles for other NTS NE neuron target regions, focusing specifically on the PVH. They show that food presentation promotes a biphasic NE response in the PVH (up and then down), while glucoprivation promotes a slightly delayed increase in PVH NE. Apparently similar response are observed when measuring the activity of nerve terminals that project from NTS Th neurons to the PVH. Furthermore, activating the PVH terminals from NTS (or rostral NTS) Th neurons increases feeding, while inhibiting these projections (but not projections from the posterior NTS) decreases feeding following a glucoprivation.

Activating PVH projections from NTS PVH neurons activates some PVH Mc4r neurons, but inhibits others, and decreases the activity of PVH Mc4r neurons *in vivo*, in a NE-dependent manner. NE agonists increase the activity of ARC Agrp neurons, and inhibit PVH Mc4r neurons *in vivo*; similar results are observed with 2DG. Activation of PVH Mc4r neurons blocks 2Dg-dependent feeding.

The authors conclude that 2DG promotes glucoprivic feeding by promoting release of NE onto PVH Mc4r neurons (and ARC Agrp neurons), thereby inhibiting PVH Mc4r neurons to promote feeding. There is reasonable (perhaps even probable), but not incontrovertible, support for this model. Lacking are experiments that block the release of NE from the NTS into the PVH and examine glucoprivic feeding (examining the effects of blocking NTS Th-dependent activation of AgRP and examine effects on the activity of PVH Mc4r neurons and glucoprivic feeding would also be interesting).

Furthermore, the activation of PVH Mc4r cells is expected to suppress food intake in response to most, if not all, stimuli, so it is difficult to interpret this particular experiment to prove that the inhibition of PVH Mc4r neurons by NTS Th neurons mediate glucoprivic feeding.

Other comments:

It would be interesting to directly compare the time courses of the effects observed in Figures 1 and 2.

Because there remains some confusion about the possibility of two distinct, and to some extent oppositely-acting, populations of NTS Th neurons, it would be interesting to know more about the effects of the posterior population- does the activation of this group of cell suppress feeding as one might expect?

Why do the authors refer to rostral NTS Th neurons in one figure, but anterior NTS Th neurons in another- are they attempting to draw a distinction?

Reviewer #3 (Remarks to the Author):

In this study, Sayar-Atasoy and colleagues report an interesting series of experiments demonstrating that adrenergic modulation of the paraventricular nucleus of the hypothalamus during feeding. The authors found that afferent from the nucleus of the solitary tract (NTS) are the major source of adrenergic tone in the PVN during feeding and that a sub-population of these NTS neurons is capable of triggering an orexigenic behavioural response. The authors further identify melanocortin receptor 4-expressing neurons in the PVN as one of the target of catecholaminergic NTS neurons.

The NTS, and the caudal brainstem in general, have long been known for their ability to trigger potent anorexigenic response, in particular in the context of satiation, meal termination, or sickness induced anorexia. Little functional information on orexigenic pathways originating from these brain structure are available. This manuscript contributes to fill this gap in knowledge, with a series of logical and well-designed experiments. Overall, the study will be of interest for the field. I have a few comments that I hope the authors will be able to clarify and few suggestions for improvement.

- It is not always easy to understand what part of the NTS is being targeted in each study. If the authors suggest that the rostral NTS-TH neurons are involved in the orexigenic response, why they did not target this NTS subregion specifically for all their experiments? Also, the narrative of the paper is confusing on this aspect, as they authors often generally refer to NTS-TH neurons. For example, what part of the NTS was targeted for the optogenetic stimulations presented in figure 3? And if no distinction was made, why the net effect of is a strong orexigenic response? I was expecting to see differential activation of rostral versus caudal NTS portions to appreciate their distinct contribution to feeding.
- It would be useful if the authors could provide a more granular analysis of the fibre photometry experiments presented in figure 1. Specifically, it is not clear whether the first peak of activity recorded from NTS-TH→PVN axon terminals is anticipatory in nature and occurs just when food is presented. Can the data be also visualised in relation to the first consummatory event? It seems that the first excitatory peak last for about 10 minutes. How the biphasic response correlate with specific phases of feeding behaviour? It would be useful if the authors could make these aspects clearer. It is also likely that the biphasic response is driven by different subpopulation of NTS-TH neurons. If they agree, could they discuss this possibility more clearly?
- Also, it would be interesting to know whether mice were food deprived during the photometry recordings and whether the recorded responses are state dependent. The methods are vague on this aspect.
- The authors state that the NTS-TH→PVN circuits is 'rapidly' activated in response to ghrelin, insulin, and 2-DG. Arguably, the responses are rather slow, in particular that to ghrelin. It is not clear what the

authors makes of these data. A discussion of how these pieces of evidence fit into the proposed model and support the authors' hypothesis could be discussed further. Do the authors think that ghrelin might shift baseline activity level of the NTS-TH→PVN circuit?

- If not necessary, I would suggest that the insulin data to be removed as they are not very convincing, most likely because of the complexity and the heterogeneity of the neuronal and endocrine response involved.
- The authors refer to the NTS-TH→PVN as a pathway conveying visceral information to the PVN. Given the rapid emergence of the response and considering that the rostral NTS integrate primarily gustatory inputs, could NE response – at least the first excitatory peak – be driven by cephalic rather than visceral signals? A discussion of these aspects would be useful.
- Do the NTS-TH neurons targeting the PVN also send collateral to other brain regions? And in particular, do they send collaterals to the arcuate nucleus (ARC)? The authors might be able gather this information from their intersectional study without additional experiments. The absence of ARC collateral would increase the strength of the elegant chemogenetic inhibition experiment where CNO was delivered systemically. This info will also increase confidence with the interpretation of axon-targeted optogenetic stimulation. In particular, I was confused when the authors indicate that there was robust somatic FOS expression in the NTS upon optogenetic stimulation of axon terminals. Is this because antidromic back propagation? If that is the case, knowing whether these neurons send collaterals to the ARC became much more important.
- The finding that the excitatory response requires both glutamate and NE, while the inhibitory response only requires NE is intriguing and is a nice further functional parcellation of NTS-TH neurons. Do the authors think that this is because of different glutamatergic and non-glutamatergic NTS-TH subpopulations? How that relates to the proposed orexigenic and anorexigenic rostro-caudal functional segregation that the authors uncovered?
- Again, in relation to phenotype of this cells, the authors indicate that activation of rostra NTS-TH neurons trigger a long-lasting orexigenic response, but do not discuss this aspect. Is this effect mediated by NPY? Could these data be integrated in the last section of the discussion where the authors discuss the involvement of additional signalling molecules? Or do the authors have alternative explanations?
- I found the discussion section describing the potential involvement of the NTS-TH→PVN circuit in stress responses and the relationship between stress responses and feeding confusing. Primarily because the authors refer to the general NTS-TH population rather than the orexigenic rostral sub-population. Was the response to stressors that the authors mentioned (as data not shown) recorded from the rostral or

caudal NTS axons? Also, given the heterogeneous nature of the stressors, again, the suggestion that the circuit convey visceral information become slightly weaker. Given these levels of ambiguity and the acknowledge complex relation between stress and appetite, it seems this section add very little to the bottom-line message of this manuscript.

Minor points.

In the introduction, the statement that 'several anti-obesity drug target the adrenergic system' is misleading as it stands. Could the authors rephrase and elaborate this concept further? Whether some of the current medications might recruit adrenergic activity, this is not a primary and/or desirable mechanisms.

Line 29. Should read 'the' paraventricular nucleus

Line 29. Should read 'that' integrates

Fibre photometry should not be referred to as an imaging technology

Figure 2C. I feel the TH staining in the PVN would be confusing for readers outside the field, since there is also (as expected) TH resident cells and co-localisation with the NTS fibre is not immediately evident, at least in the way the pictures are presented. At a first look it seems to imply that NTS-TH fibres appose to PVN-TH neurons. Is this necessary? Alternatively, a higher magnification image of a field with only fibres would work better.

It would be better to present the PCR data including NTC as part of the same gel.

With regards to the data presented in figure 5, it would make more sense to present first the in vivo recordings followed by the more stringent ex vivo circuit mapping validation. While elegant, the in vivo data are less stringent.

Response to Reviewers

We would like to thank to all reviewers for the suggestions and thorough evaluation of our manuscript, which we feel that helped to improve our manuscript substantially. Below we addressed each one of the comments along with the newly added experiments.

Reviewer #1 (Remarks to the Author):

Sayar-Atasoy and colleagues report results from a series of experiments designed to target nor/adrenergic NTS neurons that project to the PVN, and investigate their potential roles in feeding behavior after fasting and a variety of cell-specific experimental manipulations. Their reported data are generally consistent with other reports indicating that nor/adrenergic inputs to the PVN are functionally and anatomically heterogeneous, and participate in physiological and behavioral responses that underlie the central control of energy balance. Novel data reported here further implicate MC4R-expressing PVN neurons in responses to nor/adrenergic inputs that drive food intake.

*The authors interpret their experimental results in a manner that is largely appropriate, and consistent with their data. **These results should be of interest to a wide audience of researchers working on food intake and energy balance control.** However, the importance of these findings could be strengthened by addressing or clarifying several points, as indicated below:*

We thank to the review for the constructive comments and suggestions.

***RI.1a.** The first experiment includes a sophisticated in vivo assessment of extracellular NE/E within the PVN, achieved by measuring fluorescent signaling responses in Sim1 PVN neurons expressing AAV-DIO-GRABNE2h sensor. The source of AAV-DIO-GRABNE2h for these experiments should be indicated.*

This information is provided in the Methods section in Table-1 as: WZ Biosciences Catalog # YL003012

***RI.1b.** Line 99-101: what was the inedible control object, and was it novel? (same question for experiment illustrated in Fig. 2D-G). Transient NE/E signaling (and Ca²⁺ signaling) in response to different stimuli is likely modulated by physiological state (i.e., fed vs. fasted) in a manner that impacts stimulus salience during each state.*

We used an empty Eppendorf tube as 'inedible object', to which mice were previously introduced with. This was depicted in the figures and also now explicitly described to the updated manuscript text.

Evaluating modulation of the food response by physiological state is a highly informative experiment and we now performed this in the updated manuscript. For this, we measured NE/E from PVN and also axon-GCaMP activity of NTSTH terminals in PVN. Unlike fasted mice, we found that in free feeding mice there was no significant difference between presentation of chow food and inedible object, suggesting that the biphasic response profile is specific to fasted state (**Figure 1**). These results are now added to the Supplementary Fig. 1.

Figure 1. Effect of food on nor/adrenergic input to PVN depends on physiological state. PVN NE/E levels and NTSTH→PVN activity change similarly in response to food and inedible object in free feeding mice. **a** Schematic of GRAB_{NE}2h signal measurement from PVN. **b,c** Average line graph (**b**) and heat map of individual mouse (**c**) depicting change in GRAB_{NE}2h signal over time in response to chow food or inedible object presentation. **d** Average bar graph of change in GRAB_{NE}2h signal compared in response to object and chow presentation. N = 6 mice. **e-h** Same as in (a-d) except that NTSTH→PVN projection activity is measured using axon-GCaMP expression in NTSTH neurons. N = 5 mice.

RI.1c. In Fig. 1D, both the control object and food produced initial spikes in NE/E sensing by transfected Sim1 neurons, although the increase was maintained for a few more minutes before falling back towards baseline during P1 in mice that (presumably) continued to interact with and consume the food while reversing their state of deprivation. NE/E signaling then fell below the pre-P1 baseline during P2, i.e., after 10 min of food access. One might interpret the apparent decline in extracellular NE/E during P2 as a return to a lower baseline of NE/E signaling under non-fasted conditions, as discussed (lines 358-361). To support or challenge this interpretation, baseline signaling in the same mice under fasted vs. non-fasted conditions could be compared, assuming that the same mice were used for all conditions (Fig. 1; although whether the same mice were used is not explicitly stated). Similar comments apply for Ca2+

signaling in experiments producing data shown in Fig. 2. The use of Z-scores for these NE/E and Ca²⁺ measures is useful to normalize different baselines across animals, but within-subjects effects could/should be quantified. Such data could be used to support the statement (lines 176-177; 296-297) that fasting promotes NE/E signaling to PVN neurons.

We agree with the reviewer that directly comparing the signals from fed and fasted states in the same mouse would be strongly supportive of our hypothesis. The abovementioned experiments were performed in the same animals but in different sessions. Ideally, it would be preferred to compare signal levels in the same mouse and during the same recording session as the animal slowly proceed from sated to deprived state.

To address this, we performed a new set experiment in which we recorded continuously from the same mice in the presence or absence of food for 18 hours. To avoid bleaching, we adapted a protocol from Jones et al.,¹ in which, 5 minutes long measurements were made every 30 minutes and then these recordings were concatenated into one continuous string. We measured PVN NE/E levels as well as axon-GCaMP activity from NTSTH terminals in PVN. The recordings started at mid light phase (ZT6), at which mice are likely still relatively sated. Food was removed and the recordings continued for another 18 hours. We performed the same measurement in the presence or absence of food and then compared the signal at the beginning (sated) and end (fasted) of the recordings. We found that in the absence of food both PVN-NE and NTSTH→PVN AxonGCaMP signals were significantly increased, compared to the change observed under free feeding (**Figure 2**).

Along with a myriad of physiological changes, a key hallmark of hunger state is increase in AgRP neuron activity. Given that much of the hunger state physiology and behavior can be mimicked by artificial activation of AgRP neurons, we asked whether this activity could also lead

to increased NE/E release in PVN. To test this, we expressed hM3Dq activating DREADD in AgRP neurons of *Agrp-ires-cre* mice in which PVN NE/E levels were simultaneously measured by GRAB_{NE2h} sensor. We found that chemogenetic activation of AgRP neurons by DCZ ligand injection in sated mice steadily increased PVN NE/E signals but not in control mice (**Figure 3**).

Figure 3. AgRP neuron activation in sated mice elevates baseline NE/E levels. a Schematic of viral injections and fiber photometry-based NE/E measurement during chemogenetic AgRP stimulation. **b,c** Average fiber photometry trace (b) and individual heat maps (c) showing GRAB_{NE2h} signal with or without concurrent AgRP neuron stimulation. **d,e** Quantification of GRAB_{NE2h} fluorescence in AgRP:hM3D expressing (d) and control (e, no DREADD) mice after saline or DCZ ip injection. N = 5 mice/group.

The unexpected observation that systemic AgRP neuron activation elevates NE/E levels in the PVN, which in turn potentiates release from AgRP synaptic terminals, suggests a positive feedback loop. The steady increase in nor/adrenergic signals with hunger and AgRP neuron activation adds further support to the existing results with ghrelin and 2DG and suggest that this pathway can be activated by deprivation signals. We now added these new results to the manuscript in Supplementary Fig.2.

R1.1d. Lines 102-103: ghrelin and insulin are described as "rapidly" increasing PVN NE/E levels, although the data in Fig. 1 indicate that the effects occur on different and rather slow timescales. Potential bases for these different timescales should be discussed.

We agree with the reviewer, additionally, "rapidly" is a relative term, making it more problematic. Therefore, we now removed the description "rapidly". Perhaps a better description would have been "steadily"; but we left the judgement to the reader.

We think these agents, insulin, 2DG, ghrelin, are likely recruiting partially non-overlapping pathways at varying intensities and the resulting heterogeneous activation pattern may underlie the different NE/E response dynamics. For example, it was previously shown that 2DG and insulin injections cause different degree of cFos expression in lateral parabrachial nucleus with the former causing much more robust activation².

The general significance of these observations is not clear, given previous reports that systemic administration of CCK-8 (a known satiety hormone) and various stressful stimuli (including 2-DG) are known to increase extracellular levels of NE in the PVN, at least in rats. Are all of these NE/E responses to different treatments attributable to the recruitment of hindbrain NE/E neurons by "homeostatic challenges" (i.e., stress)? This is discussed indirectly in lines 332-339; more explicit links could be developed.

The Reviewer is correct that consistent with earlier findings, our own ongoing studies also suggest that this pathway is highly responsive to 'stress'. Using the live imaging approaches described in this study, we were able to replicate and expand previous cFos mapping studies to show that this pathway is activated by a wide variety of physiological and psychological stressors including hunger, hypoglycemia but also CCK, LiCl, tail pick etc.

As the Reviewer suggests, this response profile seems to be consistent with activation by 'homeostatic challenges'. We are currently actively working to understand whether various stressors activate the same set of ascending NTSTH fibers or there is stimulus specificity in recruiting distinct NTSTH subsets and their PVN targets. Although fiber photometry has a great temporal resolution, it lacks the required spatial resolution to address such issues and we are currently exploring single cell imaging approaches as an alternative. We feel that without such further insight, expanding the discussion on this section might be perceived as speculative. Indeed, comment # **R3.10** by the Reviewer-3 suggests that the existing discussion is not adding much to the manuscript unless further functional and neuroanatomical details can be provided.

To highlight these points, we now added the following sentence to the end of this section in Discussion:

"Further imaging and functional studies with increased resolution are needed to establish whether the same subsets of NTSTH neurons are recruited by various homeostatic challenges, including hunger, and how this information is conveyed to PVN neurons." (line 367)

***R1.1e.** As an overall comment regarding interpretation of results of stimulating NE/E signaling: Perhaps stimulation promotes PVN-mediated physiological outcomes (e.g., sympathetic or HPA axis activation), which then leads to increased feeding motivation via distributed neural circuits?*

This is an interesting possibility and indeed there is sufficient premise to support it, at least for HPA activation and subsequent corticosterone release, in promoting hunger³. However, several lines of evidence suggest that even if HPA activation is contributing to the feeding response, this is likely to be not the only mechanism; **(1)** our data suggest that activating NTSTH→PVN projection rapidly inhibits MC4R neurons in a NE/E dependent manner. Since MC4R neuronal inhibition itself is strongly orexigenic⁴, these neurons are likely to be among downstream effectors. Since MC4R neuronal inhibition occurred within a few hundred milliseconds after NTSTH axonal stimulation, it is highly likely that this is a local effect rather than through HPA activation. **(2)** To better understand the contribution of HPA activation, we repeated NTSTH→PVN stimulation mediated feeding experiment in the presence of the glucocorticoid receptor antagonist, mifepristone. We found that systemic administration of mifepristone (5 mg/kg) did not affect NTSTH→PVN mediated feeding response (**Figure 4**). This result is now added to the Supplementary Fig. 4d.

Figure 4. NTSTH→PVN stimulation dependent feeding does not require HPA activation.

Systemic administration of glucocorticoid receptor (GR) antagonist mifepristone did not block feeding in response to NTSTH→PVN activation (n =5).

R1.2. The terms noradrenergic and nor/adrenergic are used interchangeably throughout the manuscript. The applied models and methods (e.g., TH immunolabeling, TH-Cre mice, GRAB-NE sensing) do not discriminate noradrenergic (i.e., A2) from adrenergic (i.e., C2/C3) neurons within the NTS or their axon terminals within the PVN. For this reason, nor/adrenergic seems to be the preferred terminology.

As suggested, we corrected the terminology with nor/adrenergic throughout the text.

R1.3. With regards to the comment above (#5), published work indicates that the TH-Cre and TH-ires-Cre mice used in this study display Cre expression by neurons that do not express TH in adult mice (although they may have earlier in development). This also is evident in Fig. 2B (and S1A), although the authors describe the data as evidence for tdTom expression "specifically in catecholaminergic neuron soma" (line 122). What proportion of fluorescent reporter-labeled NTS cells are TH⁺? The authors should report these data and acknowledge that dorsal hindbrain neurons in addition to NTS nor/adrenergic neurons were likely transfected in experiments that depend on Cre expression. This does not invalidate the authors' experimental strategies or results, but interpretation of results from all experiments that depend on phenotypic specificity of Cre (and FLP) expression should be more conservative, taking the limitations of the models into consideration. For example, it would be relevant to know what proportion of mCherry⁺ neurons were TH⁺ in the experimental results shown in Fig. 4A, E; in these panels, mCherry reporter labeling seems more extensive than one might expect given the intersectional viral strategy.

This is a fair concern and transient developmental TH expression may drive adult off-target expression in reporter lines that are crossed with *Th-ires-cre* knock-in mice, which is a general limitation of some cre driver lines. In this study, we did not use any reporter line, instead, we exclusively relied on stereotaxic viral injections in adult mice, which should circumvent these limitations. Nevertheless, due to the long distance between the medulla and the hypothalamus, we typically had to wait at least ~6-8 weeks after stereotaxic injections to ensure sufficient transgene expression (ChR2, GCaMP..etc) in the NTSTH →PVN axon terminals. Therefore, we cannot rule out the possibility that transient activation of *Th* promoter during this wait time may drive transgene expression in neurons that do not visibly express TH on the experiment day.

Dynamic regulation of *tyrosine hydroxylase* promoter by physiological states is beyond the scope of this study; however, to estimate the extent of likely misexpression we injected NTS region of *Th-ires-cre* mice with AAV-FLEX-EGFP virus and quantified the portion of GFP⁺-expressing neurons that are also immunolabeled with TH staining. We found that 80% of GFP⁺-expressing neurons were also TH⁺ (**Figure 5**). Lack of TH in a small fraction of GFP⁺-expressing

neurons might be due to several reasons: (1) weak TH expression below detection threshold, (2) insufficient TH antibody penetration, (3) transient TH promoter activation and inactivation after cre-dependent viral injection. Relatedly, recent studies suggested that, under certain conditions, a small fraction of cre-dependent virus may spontaneously invert and cause leak expression^{5,6}. We think that these are general limitations inherent to cre-dependent transgene expression approach in addition to those specific to the line.

Figure 5. Majority of the cre-dependent transgene expressing neurons in *Th-ires-cre* mouse NTS are also TH+. Left: Schematic of AAV-FLEX-EGFP expression in the NTS of *Th-ires-cre* mice. Middle: Representative images showing viral EGFP expression, TH immunostaining and their overlaid image. Right: Quantification of the ratio of EGFP expressing neurons that are also immunolabeled for TH. N = 2 mice, 137 neurons, scale: 50 μ m.

Nonetheless, even if a subset of activated fibers were TH negative on the experiment day, we think this is unlikely to be consequential since our imaging and electrophysiology recordings using adrenergic receptor antagonism confirmed involvement of nor/epinephrine to NTSTH→PVN signaling.

Based on these considerations, we now added a section to the end of the Discussion to acknowledge possible transgene misexpression in a subset of TH-negative NTS neurons due to the abovementioned reasons.

“A limitation of our study is the use of Cre driver lines line to access catecholamine neurons. Due to long post-surgery wait time, we cannot rule out the possibility that a small subset of labeled neurons might be TH negative in our experiments. However, based on imaging and pharmacology experiments, which confirmed requirement of adrenergic transmission, we think that such misexpression is to be functionally inconsequential.” (line 434)

R1.4. Fig. 2I shows a consistent fall in baseline Ca²⁺ signaling from NTS axons within the PVN during the first 10min after 2DG injection. Conversely, Fig. 1H depicts less reduction of NE/E signaling below baseline within the PVN during the same timeframe after 2DG (except for a rapid decline in one mouse). The bases for this apparent timing difference in Ca²⁺ and NE/E signaling should be discussed.

We think this is likely due to adrenergic input arriving from other areas. One possibility is that ventrolateral medullary (VLM) catecholamine neurons, which are also known to be activated by hypoglycemia and project to PVN, may have higher sensitivity to glucoprivation and thus could be activated earlier than NTSTH neurons or may be differentially activated depending on rate of glucose fall⁷. This may provide the early onset NE/E to PVN even before NTSTH neurons are fully activated thereby preventing the initial drop in GRAB_{NE2h} signal. Accordingly, we now added the following section to discuss these possibilities. It is interesting to note that, in response to a comment by Reviewer-2, #R2.3 we overlaid the NE/E and axonGCaMP traces

(see Figure-9 below) and to our surprise, we noticed that NE/E slightly precedes the AxonGCaMP activity, adding further support to this explanation.

“However, unlike NE/E signal, NTSTH→PVN activity briefly dropped below baseline. It is possible that lack of a corresponding drop in NE/E signal might be due to adrenergic input from areas other than NTS which may be rapidly activated by 2DG” (line 152)

RI.5. Fig. 3A – what is the red non-nuclear labeling visible in the enlarged inset? What proportion of eYFP+ NTS neurons are TH+ after mid-NTS injection, and after anterior NTS injection? (related to point 3, above).

Following optogenetic activation of NTSTH neurons, we occasionally see cFos positive nucleus (center of which appears to be out of imaging plane in this case) even in neurons that do not express ChR2. We think that this is likely due to a local network effect. That is, activation of ChR2+ TH-neurons (which are glutamatergic) may locally activate neighboring ChR2-eYFP negative neurons. Mapping out the local connectivity of NTSTH neurons would be of interest on its own. Specificity of transgene expression in *Th-ires-cre* knock-in mice is discussed above (R1.3).

RI.6. It's not clear why control mice took 100 min to enter the feeding area (Fig. 3H), or why mice with optogenetic stimulation still took ~20 min to enter the feeding area. Were all mice pre-acclimated/familiarized with the pellet delivery system? Why the slow timescale of approach and feeding? (even in non-fasted mice, one might expect a rapid inspection of the feeding area, rather than what looks like avoidance of it, Fig. 3G, H, I).

That is correct, we performed the behavioral experiments in pre-acclimatized cages. Mice spend at least 3 days in these cages continuously, with feeders and all ports, prior to the start of stimulation experiment. Therefore, there is no need for them to inspect feeding area since it has already been there. Additionally, these experiments were performed in early light phase during which mice consume little, therefore they would have little motivation to visit food area.

Like the Reviewer, we also noticed that activating entire NTSTH→PVN projection drives feeding in a significantly slower fashion compared to activation of rostral rNTSTH→PVN projection. As the Reviewer would agree, the approach can be considered as “fast”, particularly in mice with rostral-NTSTH neurons activated (<5mins, Fig. 3k,l). Indeed, we highlighted this contrast in Fig. 3n. A likely explanation is that concurrent activation of caudal cNTSTH→PVN input diminishes the orexigenic drive of the rostral rNTSTH→PVN input.

RI.7. Fig. 3E is presented as if cell-attached recordings of NTS neurons were performed during PVN optogenetic stimulation and simultaneous food intake in the same mice, although the described ephys is performed ex vivo.

We now added a comment to the corresponding figure legend to explicitly state that the recordings in the top panel were performed from acutely prepared slices to test the faithfulness of optogenetic stimulation.

R1.8. The progressive ratio experiment (data in Fig. 3Q-S) is difficult to understand. Fasted mice (red) apparently earned 10 pellets with 10 nose pokes, on average, while mice with optogenetic stimulation of rNTS TH+ neurons earned 10 pellets with about 7 pokes, on average. This doesn't fit with the described progressive ratio strategy (lines 691-695).

This is because not every successful pellet acquisition counts toward the breakpoint measurement. Breakpoint has an added layer of stringency to perform required nose pokes for a single pellet within 30 minutes. Thus, mice may still acquire a pellet even if the total required number of nose pokes achieved in a timeframe >30 mins but this would not be added into breakpoint measurement. This condition is typical of such progressive ratio protocols to distinguish habitual nose pokes from goal directed behavior. This is highlighted in the Methods as “The break point was defined as the number of pellets where animals stopped working for more than 30 mins.”

R1.9a. Figure 4 (and related Fig. S3, in which the distribution of fluorescent reporter-labeled neurons seems quite non-specific to TH+ neurons): While the overall data support differential effects of rostral vs. caudal NTS targeting, this interpretation could be strengthened by documenting the phenotypic specificity of flp-dependent NTS neuronal transfection.

We did not perform additional TH immunolabeling and cell type characterization of transduced neurons for each experimented animal, however, as mentioned above, the specificity of expression in *Th-ires-cre* line is fairly high, but not perfect, for the likely reasons discussed in more detail in **#R1.3**. That said, the experiments in Fig.4 used an intersectional strategy such that an additional layer of misexpression can be potentially introduced by leaky flp-FRT inversion. That is, even if cre-dependent FLPO expression is highly specific in the retrogradely labeled neurons, the flp-dependent second virus (fDIO-hM4Di-mCherry) may misexpress. To test for this, we performed additional control experiments in which same intersectional injections as in Fig.4 were performed (retroAAV-DIO-FLPO in PVN and AAV-fDIO-hM4Di-mCherry in NTS) but in wt mice to evaluate the degree of flp-independent hM4Di-mCherry leak expression. To ensure correct localization of injections, a cre-independent EGFP expressing virus was added to the injection cocktail. We found that out of 5 mice, we could not detect a any off-target hM4Di-mCherry expressing soma in the NTS, suggesting that if there were any misexpression, this was likely originating from cre-lox system as discussed in R1.3.

Figure 6. Flp-FRT based hM4Di-mCherry expression has limited or no leak expression. Top left: Schematic of injections to test magnitude of leak expression in flip-dependent hM4Di-mCherry. Top right: representative image showing injection area (PVN) of rAAV-DIO-FLPo marked by coinjected AAV-GFP. Lower panels: Images from NTS showing successful viral delivery (EGFP, left) but lack of hM4Di injection (middle). N = 5 mice.

R1.9b. The effect sizes in B and C are rather small, with the experimental effect in B seemingly due to a higher baseline intake in the saline-injected control condition in hM4Di mice. In addition, the hM4Di mice seem more responsive to the hypophagic and corticosterone effects of 2DG compared to the eYFP control mice. The magnitude of feeding suppression after CNO administration seems similar between the eYFP control and hM4Di-expressing mice, and the effect of DCZ to attenuate the cort response in mice with anterior NTS hM4Di expression seems largely due to the greater cort response to 2DG in these mice (H). Effect sizes for these within-subjects data should be reported, and then compared between groups. Finally, why were different fluorescent reporter control viruses used (eYFP as control virus, but mCherry expressed with hM4Di)?

As the Reviewer predicted, rostral DREADD expressing group was more sensitive to 2DG's cort elevating effect than the control group; however, both rostral and caudal NTS groups were similarly sensitive and only the rostral one showed significant drop in cort release with DCZ. 2DG sensitivity difference was moderate in terms of feeding response. We now calculated and reported the effect sizes in the corresponding figure legend for feeding and cort responses.

Use of eYFP as opposed to mCherry was due to vector availability at the moment and to our knowledge no previous study showed different fluorophores causing distinct physiological effects.

R1.10. Fig. S1, the location of LC is incorrectly schematized

This is corrected for more accurate location.

R1.11. Fig. S3, why is mCherry labeling depicted in green channel?

This is because mCherry signal was amplified by immunolabeling using a green secondary. We now added this explanation to the corresponding figure legend.

Reviewer #2 (Remarks to the Author):

Following up on their previous observations that some Th-expressing NE neurons in the NTS promote food intake in response to fasting and glucoprivation and that projections to the ARC may explain some, but not all of this effect, the authors have now examined roles for other NTS NE neuron target regions, focusing specifically on the PVH. They show that food presentation promotes a biphasic NE response in the PVH (up and then down), while glucoprivation promotes a slightly delayed increase in PVH NE. Apparently similar response are observed when measuring the activity of nerve terminals that project from NTS Th neurons to the PVH. Furthermore, activating the PVH terminals from NTS (or rostral NTS) Th neurons increases feeding, while inhibiting these projections (but not projections from the posterior NTS) decreases feeding following a glucoprivation.

Activating PVH projections from NTS PVH neurons activates some PVH Mc4r neurons, but inhibits others, and decreases the activity of PVH Mc4r neurons in vivo, in a NE-dependent manner. NE agonists increase the activity of ARC Agrp neurons, and inhibit PVH Mc4r neurons in vivo; similar results are observed with 2DG. Activation of PVH Mc4r neurons blocks 2Dg-dependent feeding.

*The authors conclude that 2DG promotes glucoprivic feeding by promoting release of NE onto PVH Mc4r neurons (and ARC Agrp neurons), thereby inhibiting PVH Mc4r neurons to promote feeding. **There is reasonable (perhaps even probable), but not incontrovertible, support for this model.***

We thank to the Reviewer for the comments and suggestions.

R2.1. Lacking are experiments that block the release of NE from the NTS into the PVH and examine glucoprivic feeding (examining the effects of blocking NTS Th-dependent activation of AgRP and examine effects on the activity of PVH Mc4r neurons and glucoprivic feeding would also be interesting).

In the experiments outlined in Fig.4 of the manuscript, we selectively inhibited NTSTH→PVN projection using chemogenetics (hM4Di) during glucoprivic feeding. We found that suppression of this projection significantly attenuated feeding as well as cort release in response to 2DG injection (**Figure 7**). Arguably, chemogenetic activity suppression inhibited release of NE/E along with other transmitters (glutamate, NPY) from these terminals. However, taken together with electrophysiology and *in vivo* pharmacology experiments which suggested a critical role for NE/E in signaling to target neurons, impact of hM4Di-inhibition is likely, in part, due to suppressed catecholamine release.

Figure 7. Selective inhibition of rostral NTSTH→PVN projection suppressed glucoprivic feeding (from Fig.4 of the manuscript).

R2.2. Furthermore, the activation of PVH Mc4r cells is expected to suppress food intake in response to most, if not all, stimuli, so it is difficult to interpret this particular experiment to prove that the inhibition of PVH Mc4r neurons by NTS Th neurons mediate glucoprivic feeding.

We agree with the reviewer that on its own this specific experiment is not sufficient. However, four other pieces of observations adds further support this possibility **(i)** In newly added data to Fig. 7 of the manuscript, we found that 2DG strongly increases NE/E release onto PVN^{MC4R} neurons **(Figure 8)**. **(ii)** α_{1A} -AR agonist cirazoline suppresses PVN^{MC4R} activity *in vivo*. **(iii)** 2DG suppresses PVN^{MC4R} neuron activity and **(iv)** PVN^{MC4R} neuron inhibition is sufficient to promote feeding. Taken together with these results, it is likely that inhibition of PVN^{MC4R} activity likely to be contributing to 2DG induced feeding. Nevertheless, as the Reviewer suggest, ‘mediating’ is not the most parsimonious conclusion and therefore we intentionally avoided using it. We think that these experiments rather show that reduced PVN^{MC4R} neuron activity is ‘necessary’ or ‘permissive’ for, if not ‘mediating’, glucoprivic feeding.

Figure 8. Glucoprivation increases NE/E release onto PVN^{MC4R} neurons. **a** Schematic of NE/E measurements from PVN^{MC4R} neurons. **b,c** Average fiber photometry trace (b) and individual heat maps (c) showing GRAB_{NE2h} signal in response to 2DG injection. **d** Summary graph for quantification of GRAB_{NE2h} fluorescence in PVN^{MC4R} neurons. N = 4 mice *p = 0.0422.

Other comments:

R2.3. It would be interesting to directly compare the time courses of the effects observed in Figures 1 and 2.

This is an interesting suggestion, and it relates to the comment # R1.4 by the Reviewer-1. As suggested, we overlaid these recordings **(Figure 9)**. It appears that while the response profiles

of NE and axonal activity greatly parallel each other, we noticed a small lag (~2 min) in NTSTH→PVN response pattern. This is surprising since, If anything, we would expect axonal activity to precede NE signal but the FP data suggests otherwise. We do not think this is due to difference in distinct on/off kinetics of these sensors, which is in the order of milliseconds/seconds.

Figure 9. Comparative temporal dynamics of NTSTH →PVN activity and PVN NE/E levels. Overlaid fiber photometry traces of AxonCaMP fluorescence from NTSTH →PVN connection and GRAB_{NE2h} sensor showing PVN NE/E levels in response to food presentation (left) or 2DG injection (right). While overall patterns are largely in sync, the axonal activity slightly lag.

One possibility is that, as alluded in response to comment #R1.4, there might be additional sources for PVN-NE signal with higher sensitivity that NTSTH neurons causing them to be activated earlier / faster.

R2.4. Because there remains some confusion about the possibility of two distinct, and to some extent oppositely-acting, populations of NTS Th neurons, it would be interesting to know more about the effects of the posterior population- does the activation of this group of cell suppress feeding as one might expect?

We now performed this experiment and unlike stimulation of rostral-NTS, we did not observe any hunger promoting effect of caudal-NTS in light phase. Stimulation in dark phase also did not cause any feeding suppression (**Figure 10**). These results are now added to the Supplementary Fig. 5.

Figure 10. Photostimulating caudal NTSTH →PVN projection does not affect food intake. **a** Schematic depiction for activation of caudal NTSTH →PVN projection. **b,c** cumulative food intake (**b**) and average total food intake before, during and after photostimulation of cNTSTH →PVN projection (2 hours each) in free feeding mice in light phase. **d** summary bar graph of 2 hours dark onset food intake with and without concurrent cNTSTH →PVN projection stimulation (n = 7 mice).

R2.5. Why do the authors refer to rostral NTS Th neurons in one figure, but anterior NTS Th neurons in another- are they attempting to draw a distinction?

We apologize for the confusion and no, we do not claim any distinction. We now changed the entire manuscript and figures to adopt 'rostral-caudal' description.

Reviewer #3 (Remarks to the Author):

In this study, Sayar-Atasoy and colleagues report an interesting series of experiments demonstrating that adrenergic modulation of the paraventricular nucleus of the hypothalamus during feeding. The authors found that afferent from the nucleus of the solitary tract (NTS) are the major source of adrenergic tone in the PVN during feeding and that a sub-population of these NTS neurons is capable of triggering an orexigenic behavioural response. The authors further identify melanocortin receptor 4-expressing neurons in the PVN as one of the target of catecholaminergic NTS neurons.

*The NTS, and the caudal brainstem in general, have long been known for their ability to trigger potent anorexigenic response, in particular in the context of satiation, meal termination, or sickness induced anorexia. **Little functional information on orexigenic pathways originating from these brain structure are available. This manuscript contributes to fill this gap in knowledge, with a series of logical and well-designed experiments. Overall, the study will be of interest for the field. I have a few comments that I hope the authors will be able to clarify and few suggestions for improvement.***

We thank to the reviewer for the comments and suggestions.

R3.1a. It is not always easy to understand what part of the NTS is being targeted in each study.

If the authors suggest that the rostral NTS-TH neurons are involved in the orexigenic response, why they did not target this NTS subregion specifically for all their experiments?

We apologize for the confusion. Unless indicated otherwise, we targeted the entire NTS structure in all experiments. The only exceptions are Fig. 3 j-s (rostral NTSTH optostimulation), newly added Supplementary Fig. 5 (caudal NTSTH optostimulation) and DREADD inhibition experiments in Fig.4d-i.

At the beginning of this study, we did not have a priori knowledge about functional specialization of NTS subregions. Indeed, we weren't even aware that there would be a PVN projecting orexigenic population to begin with. Thus, we started our experiments by targeting the entire structure.

R3.1b. Also, the narrative of the paper is confusing on this aspect, as they authors often generally refer to NTS-TH neurons. For example, what part of the NTS was targeted for the optogenetic stimulations presented in figure 3? And if no distinction was made, why the net effect of is a strong orexigenic response? I was expecting to see differential activation of rostral versus caudal NTS portions to appreciate their distinct contribution to feeding.

Figure 3 starts with optogenetic stimulation of the axons of NTSTH neurons in PVN. These NTSTH neurons were localized across the entire rostral-caudal axis (Fig.3a-i). To better clarify this point, we now added 'Rostral+Caudal' label and updated the schematic in this figure.

What we found was that, activation of entire rostrocaudal rcNTSTH→PVN projection is orexigenic (Fig. 3a-i). When we selectively activated rostral rNTSTH→PVN projections, the response was still orexigenic and even more robust than activating axons of the entire rostral-caudal population. This is evident in the relatively higher quantity consumed and the rapid response time to initial pellet, which was significantly faster in case of rostral alone activation (Fig. 3n). As mentioned above in comment #R2.4 (Figure 10), we now added new experiments which selectively targeted the caudal cNTSTH→PVN projection (also see Supplementary Fig. 5). We found that activating caudal cNTSTH→PVN had no effect on feeding; it did not cause orexigenic response in light phase and it was also ineffective at reducing feeding at dark phase. These findings suggest that hunger signals to PVN are likely selectively conveyed by rNTSTH projections.

To place these findings in context, we and others have previously shown that globally activating entire rcNTSTH population is strongly appetite suppressing⁸⁻¹³. Taken together with our new results, we think that appetite suppressing NTSTH neurons are likely localized in the caudal NTS but their projections to PVN are insufficient to affect feeding, at least in mice^{10,12}. Notably, other projections from these neurons have been implicated for anorexigenic capacity such as parabrachial nucleus (NTS^{DbH}→PBN)⁹ and dorsomedial hypothalamus (NTSTH→DMH)¹³. It could be speculated that the rostral-caudal functional specialization among NTSTH neurons might be accompanied by distinct projections of each subpopulation; but further studies are needed to support this statement.

R3.2. It would be useful if the authors could provide a more granular analysis of the fibre photometry experiments presented in figure 1. Specifically, it is not clear whether the first peak of activity recorded from NTS-TH→PVN axon terminals is anticipatory in nature and occurs just when food is presented. Can the data be also visualised in relation to the first consummatory event? It seems that the first excitatory peak last for about 10 minutes. How the biphasic response correlate with specific phases of feeding behaviour? It would be useful if the authors could make these aspects clearer. It is also likely that the biphasic response is driven by different subpopulation of NTS-TH neurons. If they agree, could they discuss this possibility more clearly?

This point is also related to another comment below (# R3.6) i.e. whether the initial peak is due to cephalic phase anticipatory signals. A similar rise was also evident in NE/E response after food presentation (Fig. 1d). In overnight food deprived mice, feeding typically starts within the first 5 minutes after food presentation. Thus, NE/E rise might be related not only to anticipatory/cephalic signals driven by food cue but also events associated with feeding initiation and ingestion.

To distinguish between these possibilities and also to address #R3.6, we performed a new experiment in which NE/E levels in PVN were measured using GRAB_{NE}2h sensor in response to food presentation. However, this time food was inaccessible such that sensory cues would be expected to evoke 'cephalic phase' but the impact of ingestion would be excluded. We found

that similar to accessible food presentation, initial rise in GRAB_{NE2h} signal in response to inaccessible food was significantly higher than inedible object. However, unlike the response to accessible food, this signal lingered above the baseline for an extended period of time. While the NE/E signal went back down to baseline in ~10 minutes if food was available (Fig. 1d), this wasn't the case if food wasn't accessible and the response was still above baseline >40 minutes after (**Figure 11**). The prolonged NE/E signal in response to inaccessible food is likely an overlapping mixture of anticipatory cephalic response followed by stress signals. That is, inability to access food after its presentation in fasted mice could act as a stressor on its own and increase PVN NE/E signal. Consistently a recent study showed that palatable food presented in a tea restrainer significantly increased cort level and PVN activation¹⁴.

Figure 11. Initial rise in PVN nor/adrenergic signal in response to food is mediated by sensory cues. **a** Schematic of viral injection and fiber photometry-based NE/E measurement from PVN. **b,c** Average fiber photometry trace (**b**) and individual heat maps (**c**) showing GRAB_{NE2h} signal in response to inedible object (top) or inaccessible chow food (bottom). **d** Quantification of GRAB_{NE2h} fluorescence in (**b**), n = 8 mice, **p<0.01.

We agree with the reviewer that the biphasic response may originate from distinct NTSTH inputs. Specifically, as suggested from this experiment, the initial peak after food access is likely arising from a mixture of anticipatory cephalic signals and stress/arousal related signals. At least the stress/arousal related signals (particularly if the food is inaccessible) might originate from caudal NTSTH given the established role of A2 neurons¹⁵ whereas the drop in the latter phase is likely due to a decrease in the tonic activity of hunger activated rostral NTSTH input. It is important to note that, unfortunately our refeeding experiments did not extend to the point where mice are fully satiated after fasting. Thus, we may have missed the satiation phase which would have likely activated the posterior NTSTH neurons.

As discussed in **R1.1d**, these exciting results warrant further investigation, and we are currently actively studying ways to tease apart specificity and dynamics of rostral-caudal NTSTH activation patterns by different stimuli and different phases of same stimulus (e.g. feeding) using imaging approaches with higher spatial resolution.

As suggested, we added these considerations into the discussion as follows:

“Finally, activation of NTSTH→PVN pathway by metabolic cues may not be limited to visceral signals. Given the established role of rostral-NTS in integrating gustatory signals, the initial activation NTSTH→PVN projection might be mediated by cephalic phase signals. Indeed, when fasted mice were given inaccessible food, this initial rise in NE/E

levels persisted, supporting the possibility of sensory origin. Given its strong orexigenic potential, the subsequent drop in NE/E signal is likely driven by reduced tonic activity in rostral NTSTH input as feeding progresses. Since our recordings focused on the first ~40 minutes after food access, we may have overlooked satiation dependent activation of caudal NTSTH neurons, which are implicated in appetite suppression. Thus, further research is warranted to explore how rostral versus caudal NTSTH neurons participate in different phases of feeding.” (line 407)

R3.3. Also, it would be interesting to know whether mice were food deprived during the photometry recordings and whether the recorded responses are state dependent. The methods are vague on this aspect.

We now performed these experiments and evaluated the impact of metabolic state on food response in the fiber photometry recordings. We found that in *ad libitum* fed mice there was no significant difference between presentation of chow food and inedible object. For further details, please see the response to Reviewer-1 comment# **R1.1b** and new Supplementary Fig.1.

R3.4. The authors state that the NTS-TH \rightarrow PVN circuits is ‘rapidly’ activated in response to ghrelin, insulin, and 2-DG. Arguably, the responses are rather slow, in particular that to ghrelin. It is not clear what the authors makes of these data. A discussion of how these pieces of evidence fit into the proposed model and support the authors’ hypothesis could be discussed further. Do the authors think that ghrelin might shift baseline activity level of the NTS-TH \rightarrow PVN circuit?

We now removed ‘rapidly’ description from the text. To better understand the nature of NE signaling in PVN during hunger we performed additional experiments in which continuous chronic fiber photometry recordings were performed for 18 hours in the absence of food. Supporting our initial findings from ghrelin and 2DG, we found that food deprivation slowly but significantly raised PVN NE level as well as NTSTH \rightarrow PVN projection activity. Taken together, as the Reviewer suggested, we think that ghrelin, hypoglycemia and hunger elevate NE/E levels tonically in PVN. As discussed in the manuscript, these observations are in line with earlier microdialysis studies showing that ghrelin and hunger tonically increases hypothalamic NE level^{16,17}. Our findings extend these observations and show that the source of elevated NE/E originate from NTSTH neurons. We now added the following statement to the discussion to express this more explicitly.

“Our findings suggest that NTSTH neurons are likely to be the source of elevated NE/E in PVN in response to deprivation related signals.” (line 400)

As discussed in more detail in response to Reviewer-1 comment # **R1.1d**, energy deprivation is not the only stimulus activating this pathway and other ‘homeostatic challenges’ also have this capacity. This has been shown by earlier elegant cFos mapping studies as well as our own work using the imaging approach described in this study. Based on these observations, several new questions can be raised; (1) are the same subsets of NTSTH neurons activated by different homeostatic challenges? (2) What provides the specificity of behavioral response when this pathway activated by distinct homeostatic challenges? As mentioned above (#R1.1d), more refined functional imaging studies are needed to address these questions. In the Discussion subsection titled “*NTSTH \rightarrow PVN circuit as a possible integration node for stress and metabolic*

signals” we discussed the possible significance of our findings in the broader context, but we also refrained from detailing this section further for the fear of being perceived too speculative. Nevertheless, our results clearly demonstrate that food deprivation is one of those homeostatic challenges activating this circuit, moreover and artificial activation of this pathway elicit robust food seeking & consumption responses.

R3.5. If not necessary, I would suggest that the insulin data to be removed as they are not very convincing, most likely because of the complexity and the heterogeneity of the neuronal and endocrine response involved.

We removed the insulin response data as suggested.

R3.6. The authors refer to the NTS-TH→PVN as a pathway conveying visceral information to the PVN. Given the rapid emergence of the response and considering that the rostral NTS integrate primarily gustatory inputs, could NE response – at least the first excitatory peak – be driven by cephalic rather than visceral signals? A discussion of these aspects would be useful.

We thank to the reviewer for pointing this possibility. The reviewer is probably right that given the speed of initial peak and the established role of rostral NTS in conveying gustatory information, at least the first peak may very well be driven by cephalic signals. This is also supported by GRAB_{NE}2h recording experiments with inaccessible food, which showed increase in NE/E levels. As mentioned above in R3.2, we now updated the relevant section in the discussion to acknowledge this possibility as follows:

“Finally, activation of NTSTH→PVN pathway by metabolic cues may not be limited to visceral signals. Given the established role of rostral-NTS in integrating gustatory signals, the initial rapid activation phase of NTSTH→PVN projections that we observed might be mediated by cephalic phase signals. Indeed, when fasted mice were given inaccessible food, this initial rise in NE/E levels persisted, supporting the possibility of sensory origin.” (line 407)

R3.7. Do the NTS-TH neurons targeting the PVN also send collaterals to other brain regions? And in particular, do they send collaterals to the arcuate nucleus (ARC)? The authors might be able gather this information from their intersectional study without additional experiments. The absence of ARC collateral would increase the strength of the elegant chemogenetic inhibition experiment where CNO was delivered systemically. This info will also increase confidence with the interpretation of axon-targeted optogenetic stimulation. In particular, I was confused when the authors indicate that there was robust somatic FOS expression in the NTS upon optogenetic stimulation of axon terminals. Is this because antidromic back propagation? If that is the case, knowing whether these neurons send collaterals to the ARC became much more important.

The Reviewer is correct, optically induced action potentials can back propagate from PVN axon terminals and elicit cFos expression in NTSTH soma. To better understand the extent of collateralization between ARC and PVN projecting NTSTH neurons, we performed additional imaging experiments. Namely, we injected a cre-dependent retro-AAV-FLEX-GFP into the PVN of *Th-ires-cre* mice. In the same mice, we injected retro-AAV-FLEX-mCherry into the ARC. We then collected NTS sections and quantified colocalization. We found a partially overlapping

pattern such that a third (31%) of labeled neurons expressed both green and red fluorophores and the remaining 69% were either green (PVN, 34%) or red labeled (ARC, 35%).

Figure 12. A subset of PVN projecting NTSTH neurons collateralize to the ARC. **a** Schematic of multicolor retrograde AAV injections into the ARC and PVN. **b** Representative images showing successful rAAV injections that labeled local TH neurons. **c** Representative images from NTS showing retrogradely labeled EGFP (from PVN) and mCherry (from ARC) expressing NTSTH neurons. **d** Quantification of the labeled neurons in NTS. N = 3 mice, yellow arrowheads: overlapping neurons, white arrowheads: NTS neurons labeled from PVN.

Collateralization of a subset of PVN projecting NTSTH neurons to ARC suggest the possibility that the same NTSTH neurons may use redundant pathways to recruit melanocortin pathway in response to hunger signals. We think this does not invalidate or diminish our observations made in PVN for the following reasons:

- (1) In our earlier work by Aklan et al. 2020, we found that NTSTH axons activate ARC^{AgRP} neurons. Current study extends these results and conclusively show that recruitment of AgRP neurons can also occur within PVN. This is because in the slice recordings testing AgRP:Chr2→PVN connection, in which NTSTH collaterals to ARC were severed, we saw robust potentiation of GABA release from AgRP→PVN terminals in response to NE application (Fig. 6d-f). Thus, AgRP activation not only occur at both ARC and PVN, but also this activation might be mediated by the very same NTSTH neurons through collateralization.
- (2) Our GRAB_{NE}2h fiber photometry recordings by expressing the sensor either in entire PVN or specifically in PVN^{MC4R} neurons (newly added Fig. 7a-d), suggest that NE is directly released onto these neurons in response to glucoprivation, irrespective of the action of NTSTH axons in the ARC. Moreover, we also showed that α -1AR agonist rapidly suppressed PVN^{MC4R} neurons *in vivo* and this suppression is required for glucoprivic appetite.

- (3) Finally, slice recordings from PVN^{MC4R} neurons, in which ARC is not present, showed direct actions of NTSTH axon activation in PVN^{MC4R} neurons that paralleled *in vivo* fiber photometry results.

Collectively, these results suggest that while NTSTH axons can directly engage with melanocortin pathway in the PVN, a subset of the axons collateralize towards ARC and increase AgRP neuronal firing rate. This organization could help synergize the targeted orexigenic response such that while ARC projections increase AgRP neuron firing rate and suppressing POMC neurons¹⁸, projections to PVN increase release probability of ARC^{AgRP}→PVN^{MC4R} connection.

R3.8. The finding that the excitatory response requires both glutamate and NE, while the inhibitory response only requires NE is intriguing and is a nice further functional parcellation of NTS-TH neurons. Do the authors think that this is because of different glutamatergic and non-glutamatergic NTS-TH subpopulations? How that relates to the proposed orexigenic and anorexigenic rostro-caudal functional segregation that the authors uncovered?

We agree with the Reviewer that this is the probable circuit organization. We spent a great deal of effort and continue to actively work on this topic to better understand the underlying configuration. We think that distinct rostro-caudal sources of orexigenic and anorexigenic NTSTH →PVN input is likely to be only part of the story. Given that vast majority of PVN projecting NTSTH neurons are glutamatergic, additional layer of complexity is likely ensued within the PVN. That is, if all rostro-caudal NTSTH axons are equipped with glutamate and NE/E transmitters, what determines the response specificity (orexigenic vs anorexigenic)? We believe part of the answer has to do with the biphasic response profile we observed in optogenetic activation of NTSTH→PVN^{MC4R} connection, e.g. initial excitation followed by prolonged inhibition (Fig. 5l-n). Fiber photometry recordings does not reveal whether it is the same MC4R neuron that are first activated then inhibited or distinct subsets are either activated or inhibited. Although not shown here, we performed additional slice recordings to address this. We found that the biphasic response seen in fiber photometry does not extend to single neurons, i.e. a PVN^{MC4R} neuron is either activated or inhibited by mixed optostimulation of rostro-caudal NTSTH→PVN fiber bundle.

Figure 13: Possible circuit organization for rostro-caudal NTSTH→PVN^{MC4R}

Taken together, a likely circuit organization likely involves distinct sets of TH neurons originating from rostral or caudal end of NTS engage with non-overlapping sets of PVN^{MC4R} neurons causing either inhibition or excitation, respectively (**Figure 13**). We think that further work is warranted to clarify precise nature of this circuit's organization.

R3.9. Again, in relation to phenotype of this cells, the authors indicate that activation of rostra NTS-TH neurons trigger a long-lasting orexigenic response, but do not discuss this aspect. Is this effect mediated by NPY? Could these data be integrated in the last section of the discussion where the authors discuss the involvement of additional signalling molecules? Or do the authors have alternative explanations?

This is a great suggestion, and we thank to the Reviewer for it. Although we discussed NPY extensively, we missed to include its likely involvement in the long-lasting orexigenic effect, which indeed provides further support for a role for NPY. We now updated the last section of the Discussion to highlight this as follows:

“Further support for a role for NPY comes from the long lasting orexigenic effect of NTSTH→PVN stimulation (Fig. 3o), which was previously shown to be mediated by this peptide¹⁹.” (line 431)

R3.10. I found the discussion section describing the potential involvement of the NTS-TH→PVN circuit in stress responses and the relationship between stress responses and feeding confusing. Primarily because the authors refer to the general NTS-TH population rather than the orexigenic rostral sub-population. Was the response to stressors that the authors mentioned (as data not shown) recorded from the rostral or caudal NTS axons? Also, given the heterogeneous nature of the stressors, again, the suggestion that the circuit convey visceral information become slightly weaker. Given these levels of ambiguity and the acknowledge complex relation between stress and appetite, it seems this section add very little to the bottom-line message of this manuscript.

We totally agree with the Reviewer and clearly more work is needed to understand how stress and hunger signals are integrated through this pathway. However, we feel that not commenting on a possible interaction with other homeostatic challenges (stress) would be incomplete since this pathway is mostly studied in the context of stress response. Indeed, contrary to this reviewer’s statement, in the comment # R1.1d, Reviewer-1 suggests that ‘more explicit links could be developed’. Given these opposing opinions, as discussed in R1.1d, we decided to leave this part of discussion largely as is. However, we added the following sentence to highlight the need for further research to better understand stress involvement:

“Further imaging and functional studies with increased resolution are needed to establish whether the same subsets of NTSTH neurons are recruited by various homeostatic challenges, including hunger, and how this information is conveyed to PVN neurons.” (line 367)

Minor points.

R3.11. In the introduction, the statement that ‘several anti-obesity drug target the adrenergic system’ is misleading as it stands. Could the authors rephrase and elaborate this concept further? Whether some of the current medications might recruit adrenergic activity, this is not a primary and/or desirable mechanisms.

We rephrased this sentence to highlight possible secondary/unintended nature of this interaction as follows:

” Norepinephrine (NE) has long been known to robustly regulate appetite²⁰ and some of the current anti-obesity drugs are thought to recruit nor/adrenergic system as a secondary effect, that may contribute to their primary anorexigenic mechanisms²¹.” (line 57)

R3.12. Line 29. Should read 'the' paraventricular nucleus

Corrected as suggested.

R3.13. Line 29. Should read 'that' integrates

Corrected as suggested.

R3.13. Fibre photometry should not be referred to as an imaging technology

We changed "imaging" with "recording" throughout the manuscript.

R3.14. Figure 2C. I feel the TH staining in the PVN would be confusing for readers outside the field, since there is also (as expected) TH resident cells and co-localisation with the NTS fibre is not immediately evident, at least in the way the pictures are presented. At a first look it seems to imply that NTS-TH fibres appose to PVN-TH neurons. Is this necessary? Alternatively, a higher magnification image of a field with only fibres would work better.

We thank to the reviewer for the suggestion. To avoid possible confusion, we added a note to the corresponding figure legend to draw attention to the presence of local TH-expressing neurons in PVN.

R3.15. It would be better to present the PCR data including NTC as part of the same gel.

When we were performing these experiments, we tested for a number other PVN target cell types thus we had several additional lanes for other candidate transcripts. To avoid distraction from these unrelated bands we focused on MC4R-related part.

R3.16. With regards to the data presented in figure 5, it would make more sense to present first the in vivo recordings followed by the more stringent ex vivo circuit mapping validation. While elegant, the in vivo data are less stringent.

As suggested, we changed the order of panels such that ex vivo circuit mapping comes after in vivo recordings.

References

- 1 Jones, J. R., Chaturvedi, S., Granados-Fuentes, D. & Herzog, E. D. Circadian neurons in the paraventricular nucleus entrain and sustain daily rhythms in glucocorticoids. *Nat Commun* **12**, 5763, doi:10.1038/s41467-021-25959-9 (2021).
- 2 Garfield, A. S. *et al.* A parabrachial-hypothalamic cholecystokinin neurocircuit controls counterregulatory responses to hypoglycemia. *Cell Metab* **20**, 1030-1037, doi:10.1016/j.cmet.2014.11.006 (2014).

- 3 Perry, R. J. *et al.* Leptin's hunger-suppressing effects are mediated by the hypothalamic-pituitary-adrenocortical axis in rodents. *Proc Natl Acad Sci U S A* **116**, 13670-13679, doi:10.1073/pnas.1901795116 (2019).
- 4 Garfield, A. S. *et al.* A neural basis for melanocortin-4 receptor-regulated appetite. *Nat Neurosci* **18**, 863-871, doi:10.1038/nn.4011 (2015).
- 5 Fischer, K. B., Collins, H. K. & Callaway, E. M. Sources of off-target expression from recombinase-dependent AAV vectors and mitigation with cross-over insensitive ATG-out vectors. *Proc Natl Acad Sci U S A* **116**, 27001-27010, doi:10.1073/pnas.1915974116 (2019).
- 6 Botterill, J. J. *et al.* Off-Target Expression of Cre-Dependent Adeno-Associated Viruses in Wild-Type C57BL/6J Mice. *eNeuro* **8**, doi:10.1523/ENEURO.0363-21.2021 (2021).
- 7 Jokiahho, A. J., Donovan, C. M. & Watts, A. G. The rate of fall of blood glucose determines the necessity of forebrain-projecting catecholaminergic neurons for male rat sympathoadrenal responses. *Diabetes* **63**, 2854-2865, doi:10.2337/db13-1753 (2014).
- 8 Aklan, I. *et al.* NTS Catecholamine Neurons Mediate Hypoglycemic Hunger via Medial Hypothalamic Feeding Pathways. *Cell Metab* **31**, 313-326 e315, doi:10.1016/j.cmet.2019.11.016 (2020).
- 9 Roman, C. W., Derkach, V. A. & Palmiter, R. D. Genetically and functionally defined NTS to PBN brain circuits mediating anorexia. *Nat Commun* **7**, 11905, doi:10.1038/ncomms11905 (2016).
- 10 Chen, J. *et al.* A Vagal-NTS Neural Pathway that Stimulates Feeding. *Curr Biol* **30**, 3986-3998 e3985, doi:10.1016/j.cub.2020.07.084 (2020).
- 11 Kreisler, A. D., Davis, E. A. & Rinaman, L. Differential activation of chemically identified neurons in the caudal nucleus of the solitary tract in non-entrained rats after intake of satiating vs. non-satiating meals. *Physiol Behav* **136**, 47-54, doi:10.1016/j.physbeh.2014.01.015 (2014).
- 12 Murphy, S. *et al.* Nucleus of the solitary tract A2 neurons control feeding behaviors via projections to the paraventricular hypothalamus. *Neuropsychopharmacology* **48**, 351-361, doi:10.1038/s41386-022-01448-5 (2023).
- 13 Tsang, A. H., Nuzzaci, D., Darwish, T., Samudrala, H. & Blouet, C. Nutrient sensing in the nucleus of the solitary tract mediates non-aversive suppression of feeding via inhibition of AgRP neurons. *Mol Metab* **42**, 101070, doi:10.1016/j.molmet.2020.101070 (2020).
- 14 R, G. A. *et al.* A paraventricular thalamus to insular cortex glutamatergic projection gates "emotional" stress-induced binge eating in females. *Neuropsychopharmacology*, doi:10.1038/s41386-023-01665-6 (2023).
- 15 Rinaman, L. Hindbrain noradrenergic A2 neurons: diverse roles in autonomic, endocrine, cognitive, and behavioral functions. *Am J Physiol Regul Integr Comp Physiol* **300**, R222-235, doi:10.1152/ajpregu.00556.2010 (2011).
- 16 Date, Y. *et al.* Peripheral ghrelin transmits orexigenic signals through the noradrenergic pathway from the hindbrain to the hypothalamus. *Cell Metab* **4**, 323-331, doi:10.1016/j.cmet.2006.09.004 (2006).
- 17 Stanley, B. G., Schwartz, D. H., Hernandez, L., Hoebel, B. G. & Leibowitz, S. F. Patterns of extracellular norepinephrine in the paraventricular hypothalamus: relationship to circadian rhythm and deprivation-induced eating behavior. *Life Sci* **45**, 275-282, doi:10.1016/0024-3205(89)90136-7 (1989).
- 18 Aklan, I. *et al.* NTS Catecholamine Neurons Mediate Hypoglycemic Hunger via Medial Hypothalamic Feeding Pathways. *Cell Metab*, doi:10.1016/j.cmet.2019.11.016 (2019).
- 19 Chen, Y. *et al.* Sustained NPY signaling enables AgRP neurons to drive feeding. *Elife* **8**, doi:10.7554/eLife.46348 (2019).
- 20 Wellman, P. J. Norepinephrine and the control of food intake. *Nutrition* **16**, 837-842 (2000).

- 21 Muller, T. D., Bluher, M., Tschop, M. H. & DiMarchi, R. D. Anti-obesity drug discovery: advances and challenges. *Nat Rev Drug Discov* **21**, 201-223, doi:10.1038/s41573-021-00337-8 (2022).

REVIEWERS' COMMENTS

Reviewer #1 (Remarks to the Author):

Sayar-Atasoy and colleagues have provided careful, in-depth responses to most of my original comments and questions. New experiments have been added (i.e., examining fed/fasted state) to enhance interpretive strength. Overall I find this to be an excellent study, and I don't have any remaining major concerns. However, the following relatively minor points should be addressed:

1) In response to R1.1e, what about potential effects of autonomic (sympathetic) activation? The authors' response was limited to HPA axis activation.

2) In response to R1.91, the authors point to labeling in Fig. 6 (Flp-FRT based hM4Di-mCherry expression has limited or no leak expression). If I understand correctly, a non-Cre-dependent AAV-EGFP was injected into PVN (top), and also into NTS, to tag injection site placements. The labeling in the bottom row shows green neurons within the hypoglossal motor nucleus (and some in the AP), but not within the NTS. Thus, it's not clear that the "NTS" injection was accurate, making the lack of mCherry labeling a possible false-negative rather than evidence against leak expression.

3) In response to R1.11 -- Figure S3 -- I cannot find the explanation of using a green secondary to enhance mCherry signal in a corresponding figure legend.

Reviewer #2 (Remarks to the Author):

The authors have generally done a good job addressing most of the comments, with the following provisos:

Because they have not formally tested whether is the the NE release by NTS TH neurons in the PVH that increases feeding/inhibits Mc4R neurons, they should spell this out in the text- presumably in the discussion.

The authors should note in the results the lag between E/NE release and increased firing (Ca²⁺ influx) in the PVH-projecting axons of NTS TH neurons, and discuss the potential mechanisms and implications of this finding.

Reviewer #3 (Remarks to the Author):

The authors were very responsive to the suggestions put forth by the reviewers.

They incorporated supplementary experimental evidence, conducted control experiments, and offered valid conceptual clarifications. As a result, they have significantly enhanced an already well-executed and informative study.

I am satisfied with how the authors addressed my questions, as well as those from the other reviewers, and please if these have contributed to strenght the manuscript.

Consequently, I endorse the publication of this study without any reservations.

I anticipate that it will be extremely well received and attract substantial citations.

REVIEWERS' COMMENTS

Reviewer #1 (Remarks to the Author):

Sayar-Atasoy and colleagues have provided careful, in-depth responses to most of my original comments and questions. New experiments have been added (i.e., examining fed/fasted state) to enhance interpretive strength. Overall I find this to be an excellent study, and I don't have any remaining major concerns. However, the following relatively minor points should be addressed:

We thank to the reviewer for the suggestions in both rounds of revisions. Below we addressed the remaining concerns:

1) In response to R1.1e, what about potential effects of autonomic (sympathetic) activation? The authors' response was limited to HPA axis activation.

The reviewer is correct that our cort-antagonism experiment does not rule out possible contribution of autonomic activation. To acknowledge this point, we added a sentence to the Results section where the cort-antagonism experiment was described as follows:

“Notably, cort release was not required for the feeding effect since systemic administration of glucocorticoid receptor antagonist mifepristone did not affect NTSTH→PVN mediated feeding (Supplementary Fig. 4d); however, we cannot rule out possible contribution of sympathetic activation.” (line 198)

2) In response to R1.91, the authors point to labeling in Fig. 6 (Flp-FRT based hm4Di-mCherry expression has limited or no leak expression). If I understand correctly, a non-Cre-dependent AAV-EGFP was injected into PVN (top), and also into NTS, to tag injection site placements. The labeling in the bottom row shows green neurons within the hypoglossal motor nucleus (and some in the AP), but not within the NTS. Thus, it's not clear that the "NTS" injection was accurate, making the lack of mCherry labeling a possible false-negative rather than evidence against leak expression.

We apologize for the confusion. As the reviewer noted, GFP-marker expression is visible in both dorsal and ventral structures relative to NTS. This suggests that non-cre dependent GFP-expressing virus spread to these areas therefore it is unlikely that NTS, which is in the middle of them did not receive any virus. We think that the dark appearance has more to do with the extreme brightness of neighboring neurons, particularly those large soma located in hypoglossal nucleus, which reduced the relative brightness in the NTS in order to avoid saturation. Indeed, after taking a closer look into NTS in this and adjacent sections we can clearly see GFP expressing soma (**Figure 1**).

Figure 1. Brightness adjusted images from sections containing NTS prepared from non-cre-dependent GFP and flp-dependent hm4d-mCherry virus injected control mice.

3) In response to R1.11 -- Figure S3 -- I cannot find the explanation of using a green secondary to enhance mCherry signal in a corresponding figure legend.

We believe the reviewer is referring the following comment they made in the first round of revision:

“R1.11. Fig. S3, why is mCherry labeling depicted in green channel?”

Our response was as follows:

“This is because mCherry signal was amplified by immunolabeling using a green secondary. We now added this explanation to the corresponding figure legend.”

We apologize for not clarifying in our previous response that in the updated version of the manuscript, Figure S3 is now the new **Figure S6** (due to inclusion of new supplemental data in the first revision). Therefore, the reviewer likely has looked into the wrong figure legend. The new Figure S6 legend has this explanation as shown below.

“Supplementary Figure 6. Distribution of hM4Di expression in rostral and caudal NTSTH neurons.

a Schematic and representative images of hM4Di targeted to rostral NTSTH neurons. b Distribution of rostral NTS targeted hM4Di expression throughout the rostral-caudal axis of the NTS (mCherry signal was amplified with immunolabeling using a 488 secondary Ab). c Schematic for hM4Di injections into the caudal NTS (scale bars: 500 μm). d Distribution profile of hM4Di targeted to caudal NTS. Relative expression is indicated by color: dark red – high expression, light red – low expression, white – no expression.”

Reviewer #2 (Remarks to the Author):

The authors have generally done a good job addressing most of the comments, with the following provisos:

We thank to the reviewer for the constructive comments and suggestions.

Because they have not formally tested whether is the the NE release by NTS TH neurons in the PVH that increases feeding/inhibits Mc4R neurons, they should spell this out in the text- presumably in the discussion.

As suggested, we added this consideration to the paragraph starting with “A limitation of our study” of the Discussion as follows:

“We also did not directly test whether the NE/E release by NTSTH axons in the PVN increases feeding and inhibits PVN^{MC4R} neurons. Future experiments with NTSTH specific inactivation of NE/E synthesis will help clarify this.” (line 468)

The authors should note in the results the lag between E/NE release and increased firing (Ca²⁺ influx) in the PVH-projecting axons of NTS TH neurons, and discuss the potential mechanisms and implications of this finding.

As the reviewer suggested, we added these considerations to the part of the Results where we compare the two types of signals as follows:

“Additionally, we noticed that NE/E based GRAB_{NE2h} signal slightly precedes NTSTH axonal Ca²⁺ signal. It is possible that lack of a corresponding drop in NE/E signal and temporal lag between the two signals might be due to adrenergic input from areas other than NTS which may be rapidly activated by 2DG.” (line 155)

Reviewer #3 (Remarks to the Author):

The authors were very responsive to the suggestions put forth by the reviewers. They incorporated supplementary experimental evidence, conducted control experiments, and offered valid conceptual clarifications. As a result, they have significantly enhanced an already well-executed and informative study.

I am satisfied with how the authors addressed my questions, as well as those from the other reviewers, and please if these have contributed to strenght the manuscript.

Consequently, I endorse the publication of this study without any reservations.

I anticipate that it will be extremely well received and attract substantial citations.

We thank to the reviewer for the comments and thorough evaluation of our manuscript.